# Neural Polarizer: A Lightweight and Effective Backdoor Defense via Purifying Poisoned Features

**Mingli Zhu**[1][*]   **Shaokui Wei**[1][*]   **Hongyuan Zha**[1,2]   **Baoyuan Wu**[1][†]

[1]School of Data Science,
The Chinese University of Hong Kong, Shenzhen (CUHK-Shenzhen), China
[2]Shenzhen Institute of Artificial Intelligence and Robotics for Society, China

## Abstract

Recent studies have demonstrated the susceptibility of deep neural networks to backdoor attacks. Given a backdoored model, its prediction of a poisoned sample with trigger will be dominated by the trigger information, though trigger information and benign information coexist. Inspired by the mechanism of the optical polarizer that a polarizer could pass light waves with particular polarizations while filtering light waves with other polarizations, we propose a novel backdoor defense method by inserting a learnable neural polarizer into the backdoored model as an intermediate layer, in order to purify the poisoned sample via filtering trigger information while maintaining benign information. The neural polarizer is instantiated as one lightweight linear transformation layer, which is learned through solving a well designed bi-level optimization problem, based on a limited clean dataset. Compared to other fine-tuning-based defense methods which often adjust all parameters of the backdoored model, the proposed method only needs to learn one additional layer, such that it is more efficient and requires less clean data. Extensive experiments demonstrate the effectiveness and efficiency of our method in removing backdoors across various neural network architectures and datasets, especially in the case of very limited clean data. Codes are available at https://github.com/SCLBD/BackdoorBench (PyTorch) and https://github.com/JulieCarlon/NPD-MindSpore (MindSpore).

## 1   Introduction

Several studies have revealed the vulnerabilities of deep neural networks (DNNs) to various types of attacks [15, 17, 21, 22, 40], of which backdoor attacks [6, 9, 11, 36] are attracting increasing attention. In backdoor attacks, the adversary could produce a backdoored DNN model through manipulating the training dataset [5, 24] or the training process [20, 29], such that the backdoored model predicts any poisoned sample with particular triggers to the predetermined target label, while behaves normally on benign samples. Backdoor attacks can arise from various sources, such as training based on a poisoned dataset, or utilizing third-party platforms for model training, or downloading backdoored models from untrusted third-party providers. These scenarios significantly elevate the threat of backdoor attacks to DNNs' applications, and meanwhile highlight the importance of defending against backdoor attacks.

Several seminal backdoor defense methods have been developed, mainly including 1) in-training approaches, which aim to train a secure model based on a poisoned dataset through well-designed training algorithms or objective functions, such as DBD [14] and D-ST [4]. 2) post-training approaches, which aim to mitigate the backdoor effect from a backdoored model through adjusting the

---

[*]Equal contribution

[†]Corresponds to Baoyuan Wu (wubaoyuan@cuhk.edu.cn).

37th Conference on Neural Information Processing Systems (NeurIPS 2023).

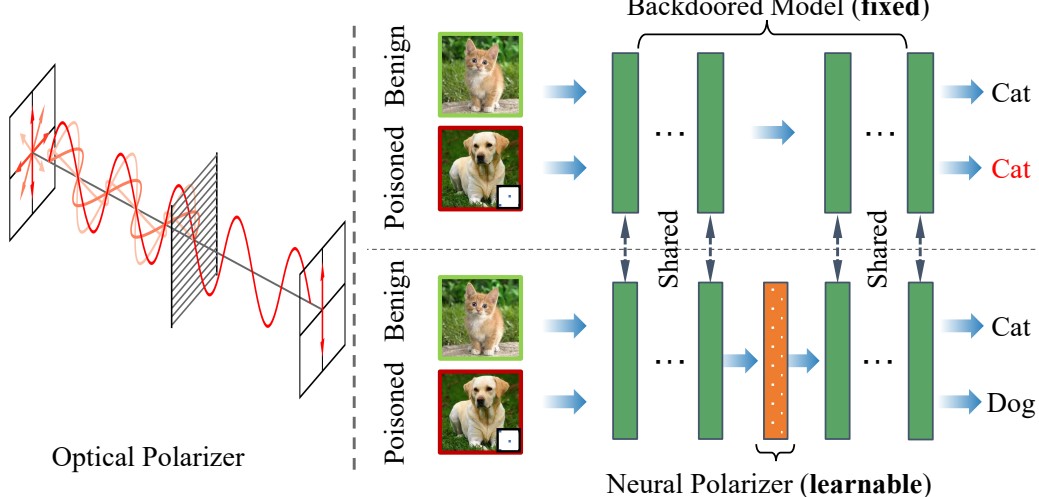

Figure 1: **Left**: Illustration of an optical polarizer [41]. Only light waves with specific polarizations can pass through the polarizer among the three incident light waves. **Right**: Defense against backdoors by integrating a trainable neural polarizer into the compromised model. The neural polarizer effectively filters out backdoor-related features, effectively eliminating the backdoor.

model parameters (*e.g.*, fine-tuning or pruning), usually based on a limited subset of clean training dataset, such as fine-pruning [23], ANP [43], or i-BAU [45]. This work focuses on the latter one. However, there are two limitations to existing post-training approaches. First, given very limited clean data, it is challenging to find a good checkpoint to simultaneously achieve backdoor mitigation and benign accuracy maintenance from the high-dimensional loss landscape of a complex model. Second, adjusting all parameters of a complex model is costly.

To tackle the above limitations, we propose a lightweight and effective post-training defense approach, which only learns one additional layer, while fixing all layers of the original backdoored model. It is inspired by the mechanism of the optical polarizer [44] that in a mixed light wave with diverse polarizations, only the light wave with some particular polarizations could pass the polarizer, while those with other polarizations are blocked (see Fig. 1-left). Correspondingly, by treating one poisoned sample as the mixture of trigger feature and benign feature, we define a neural polarizer and insert it into the backdoored model as one additional intermediate layer (see Fig. 1-right) in order to filter trigger feature and maintain benign feature, such that poisoned samples could be purified to mitigate backdoor effect, while benign samples are not significantly influenced.

In practice, to achieve an effective neural polarizer, it should be learned to weaken the correlation between the trigger and the target label while keeping the mapping from benign samples to their ground-truth labels. However, the defender only has a limited clean dataset, while neither trigger nor target label is accessible. To tackle it, we propose a bi-level optimization, where a sample-specific target label is dynamically estimated according to the output confidence, and the trigger is approximated by the targeted adversarial perturbation. Besides, in our experiments, the neural polarizer is implemented by a linear transformation (*i.e.*, the combination of one $1 \times 1$ convolutional layer and one batch normalization layer). Consequently, it can be efficiently and effectively learned with very limited clean data to achieve good defense performance, which is verified by extensive experiments on various model architectures and datasets.

Our main contributions are three-fold. **1)** We propose an innovative backdoor defense approach that only learns one additional linear transformation called neural polarizer while all parameters of the backdoored model are fixed, such that it just requires very low computational cost and very limited clean data. **2)** A bi-level optimization problem and an effective learning algorithm are provided to optimize the parameter of the neural polarizer. **3)** Extensive experimental results demonstrate the superiority of the proposed method on various networks and datasets, in terms of both efficiency and effectiveness.

## 2 Related work

**Backdoor attacks.** Traditional backdoor attacks are additive attacks that modify a small fraction of training samples by patching a pre-defined pattern and assigning them to targeted labels [8, 11]. These modified samples, along with the unaffected samples, constitute a *poisoned dataset* [11]. The model trained with this dataset will be implanted with backdoors that predict the targeted label when triggered by the injected patterns while maintaining normal behavior on clean samples [5, 50]. Recently, advanced attacks have considered more invisible trigger injection methods such as training an auto-encoder feature embedding or using a local transformation function [20, 29, 46]. To increase the stealthiness of attacks, clean-label attacks succeed by obfuscating image subject information and establishing a correlation between triggers and targeted labels without modifying the labels of poisoned samples [1, 2, 30].

**Backdoor defense.** Backdoor defense methods can be broadly categorized into training-stage defenses and post-processing defenses. Training-stage defenses assume access to a poisoned dataset for model training [10]. The different behaviors between the poisoned and clean samples can be leveraged to identify suspicious instances, such as sensitivity to transformation [4] and clustering phenomenon in feature space [14]. Most defense methods belong to post-processing defenses [39], which assume that the defender only has access to a suspicious DNN model and a few clean samples. Therefore, they must remove the backdoor threat with limited resources. There are mainly three types of defense strategies: trigger reversion methods try to recover the most possible triggers and utilize the potential triggers to fine-tune the model [35]; pruning-based methods aim at locating the neurons that are most related to backdoors and pruning them [23, 43, 47, 48]; and fine-tuning based defenses leverage clean samples to rectify the model [19, 45]. I-BAU [45] is most closely related to our method, which formulates a minimax optimization framework to train the network with samples under universal adversarial perturbations. However, our method differs from I-BAU in that our approach does not require training the entire network, and our perturbation generation mechanism is distinct. Other methods proposed for backdoor detection include Beatrix [26], which uses Gram matrices to identify poisoned samples; and AEVA [12], which detects backdoor models by adversarial extreme value analysis. In this study, we focus on post-processing defenses and primarily compare our method with state-of-the-art post-processing defenses [19, 23, 35].

## 3 Methodology

### 3.1 Basic settings

**Notations.** We consider a classification problem with $K$ classes ($K \geq 2$). Let $\boldsymbol{x} \in \mathcal{X} \subset \mathbb{R}^d$ be a $d$-dimensional input sample, and its ground truth label is denoted as $y \in \mathcal{Y} = \{1, \ldots, K\}$. Then, a $L$ layers deep neural network $f : \mathcal{X} \times \mathcal{W} \to \mathbb{R}^K$ parameterized by $\boldsymbol{w} \in \mathcal{W}$ is defined as:

$$f(\boldsymbol{x}; \boldsymbol{w}) = f_{\boldsymbol{w}_L}^{(L)} \circ f_{\boldsymbol{w}_{L-1}}^{(L-1)} \circ \cdots \circ f_{\boldsymbol{w}_1}^{(1)}(\boldsymbol{x}), \tag{1}$$

where $f_{\boldsymbol{w}_l}^{(l)}$ is the function (*e.g.*, convolution layer) with parameter $\boldsymbol{w}_l$ in the $l^{\text{th}}$ layer with $1 \leq l \leq L$. For simplicity, we denote $f(\boldsymbol{x}; \boldsymbol{w})$ as $f_{\boldsymbol{w}}(\boldsymbol{x})$ or $f_{\boldsymbol{w}}$. Given input $\boldsymbol{x}$, the predicted label of $\boldsymbol{x}$ is given by $\arg\max_k f_k(\boldsymbol{x}; \boldsymbol{w}), k = 1, \ldots, K$, where $f_k(\boldsymbol{x}; \boldsymbol{w})$ is the logit of the $k^{\text{th}}$ class.

**Threat model.** We assume that the adversary could produce the backdoored model $f_{\boldsymbol{w}}$ through manipulating training data or training process, such that $f_{\boldsymbol{w}}$ performs well on benign sample $\boldsymbol{x}$ (*i.e.*, $f_{\boldsymbol{w}}(\boldsymbol{x}) = y$), and predicts poisoned sample $\boldsymbol{x}_\Delta = r(\boldsymbol{x}, \Delta)$ to the target label $T$, with $\Delta$ indicating the trigger and $r(\cdot, \cdot)$ being the fusion function of $\boldsymbol{x}$ and $\Delta$. Considering that the adversary may set multiple targets, we use $T_i$ to denote the target label for $\boldsymbol{x}_i$.

**Defender's goal.** Assume that the defender has access to the backdoored model $f_{\boldsymbol{w}}$ and a limited set of benign training data, denoted as $\mathcal{D}_{bn} = \{(\boldsymbol{x}_i, y_i)\}_{i=1}^N$. The defender's goal is to obtain a new model $\hat{f}$ based on $f_{\boldsymbol{w}}$ and $\mathcal{D}_{bn}$, such that the backdoor effect will be mitigated and the benign performance is maintained in $\hat{f}$.

## 3.2 Neural polarizer for DNNs

**Neural polarizer.** We propose the neural polarizer to purify poisoned sample in the feature space. Formally, it is instantiated as a lightweight linear transformation $g_{\boldsymbol{\theta}}$, parameterized with $\boldsymbol{\theta}$. As shown in Fig. 1, $g_{\boldsymbol{\theta}}$ is inserted into the neural network $f_{\boldsymbol{w}}$ at a specific immediate layer, to obtain a combined network $f_{\boldsymbol{w},\boldsymbol{\theta}}$. For clarity, we denote $f_{\boldsymbol{w},\boldsymbol{\theta}}$ as $\hat{f}_{\boldsymbol{\theta}}$, since $\boldsymbol{w}$ is fixed.

A desired neural polarizer should have the following three properties:

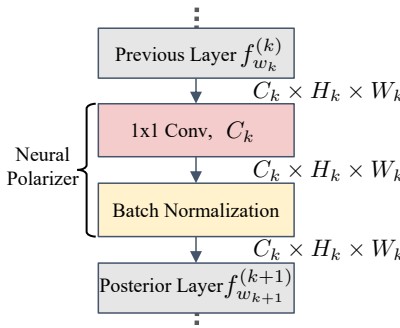

- **Compatible with the neighboring layers**: in other words, its input feature and its output activation must have the same shape. This requirement can be fulfilled through careful architectural design.

- **Filtering trigger features in poisoned samples**: after the neural polarizer, the trigger features should be filtered, such that the backdoor is deactivated, *i.e.*, $\hat{f}_{\boldsymbol{\theta}}(\boldsymbol{x}_{\Delta}) \neq T$.

- **Preserving benign features in poisoned and benign samples**: the neural polarizer should preserve benign features, such that $\hat{f}_{\boldsymbol{\theta}}$ performs well on both poisoned and benign samples, *i.e.*, $\hat{f}_{\boldsymbol{\theta}}(\boldsymbol{x}_{\Delta}) = \hat{f}_{\boldsymbol{\theta}}(\boldsymbol{x}) = y$.

Figure 2: An example of neural polarizer for a DNN.

The first property could be easily satisfied by designing neural polarizer's architecture. For example, as illustrated in Fig. 2, given the input feature map with the shape $C_k \times H_k \times W_k$, the neural polarizer is implemented by a convolution layer (Conv) with $C_k$ convolution filters of shape $1 \times 1$, followed by a Batch Normalization (BN) layer. The Conv-BN block can be seen as a linear transformation layer. To satisfy the latter two properties, $\boldsymbol{\theta}$ should be learned by solving some well designed optimization, of which the details are presented in Section 3.3.

## 3.3 Learning neural polarizer

### 3.3.1 Loss functions in oracle setting

To learn a good neural polarizer $g_{\boldsymbol{\theta}}$, we first consider an oracle setting where the trigger $\Delta$ and target label $T$ are given. We present some loss functions to encourage $g_{\boldsymbol{\theta}}$ to satisfy the latter two properties mentioned above, as follows:

- **Loss for filtering trigger features in poisoned samples.** Given trigger $\Delta$ and target label $T$, filtering trigger features can be implemented by weakening the connection between $\Delta$ and $T$ with the following loss:

$$\mathcal{L}_{asr}(\boldsymbol{x}, y, \Delta, T; \boldsymbol{\theta}) = -\log(1 - s_T(\boldsymbol{x}_{\Delta}; \boldsymbol{\theta})), \tag{2}$$

where $s_T(\boldsymbol{x}_{\Delta}; \theta)$ is the softmax probability of predicting $\boldsymbol{x}_{\Delta}$ to label $T$ by $\hat{f}_{\boldsymbol{\theta}}$. By reducing $\mathcal{L}_{asr}$, the attack success rate of the backdoor attack can be decreased.

- **Loss for maintaining benign features in poisoned samples.** To purify the poisoned sample such that it can be classified to the true label, we leverage the boosted cross entropy defined in [37]:

$$\mathcal{L}_{bce}(\boldsymbol{x}, y, \Delta; \boldsymbol{\theta}) = -\log(s_y(\boldsymbol{x}_{\Delta}; \boldsymbol{\theta})) - \log\left(1 - \max_{k \neq y} s_k(\boldsymbol{x}_{\Delta}; \boldsymbol{\theta})\right). \tag{3}$$

- **Loss for maintaining benign features in benign samples.** To preserve benign features in benign sample, we adopt the widely used cross-entropy loss, *i.e.*, $\mathcal{L}_{bn}(\boldsymbol{x}, y; \boldsymbol{\theta}) = \mathcal{L}_{\text{CE}}(\hat{f}_{\boldsymbol{\theta}}(\boldsymbol{x}), y)$.

### 3.3.2 Approximation & optimization

**Approximating $T$ and $\Delta$.** Note that as $T$ and $\Delta$ are inaccessible in both $\mathcal{L}_{asr}$ and $\mathcal{L}_{bce}$, these two losses cannot be directly optimized. Thus, we have to approximate $T$ and $\Delta$. In terms of

approximating $T$, although some methods have been developed to detect target class of backdoored models [12, 26], here we adopt a simple sample-specific and dynamic strategy:

$$T \approx y' = \arg\max_{k \neq y} \hat{f}_k(\boldsymbol{x}; \boldsymbol{\theta}). \tag{4}$$

There are two main advantages of this strategy. First, the predicted target label enables us to generate targeted adversarial perturbation. As analyzed later, the targeted adversarial perturbation is a better surrogate for the unknown trigger than the untargeted adversarial perturbation. Second, the sample-specific target label prediction is applicable to both all2one (one trigger, one target class) and all2all (every class is target class) attack settings. In practice, defenders lack certainty about attack types (all2one, all2all, or multi-trigger multi-target), risking sub-optimal defenses due to erroneous guesses or detection. Our sample-specific target label prediction avoids this risk.

In terms of approximating $\Delta$, it is approximated by the targeted adversarial perturbation of $f_{\boldsymbol{w}}$, *i.e.*,

$$\Delta \approx \boldsymbol{\delta}^* = \arg\min_{\|\boldsymbol{\delta}\|_p \leq \rho} \mathcal{L}_{\text{CE}}\left(\hat{f}_{\boldsymbol{\theta}}(\boldsymbol{x} + \boldsymbol{\delta}), y'\right), \tag{5}$$

where $\|\cdot\|_p$ is the $L_p$ norm, $\rho$ is the budget for perturbations.

**Bi-level optimization problem.** By employing the approximations of $T$ and $\Delta$ shown in Eq. (4) and (5), respectively, and substituting these approximations into the respective loss functions of Eq. (2) and (3), we introduce the following optimization problem to learn $\boldsymbol{\theta}$ based on the clean data $\mathcal{D}_{bn}$:

$$\min_{\boldsymbol{\theta}} \quad \frac{1}{N} \sum_{i=1}^{N} \lambda_1 \mathcal{L}_{bn}(\boldsymbol{x}_i, y_i; \boldsymbol{\theta}) + \lambda_2 \mathcal{L}_{asr}(\boldsymbol{x}_i, y_i, \boldsymbol{\delta}_i^*, y_i'; \boldsymbol{\theta}) + \lambda_3 \mathcal{L}_{bce}(\boldsymbol{x}_i, y_i, \boldsymbol{\delta}_i^*; \boldsymbol{\theta}),$$

$$\text{s.t.} \quad \boldsymbol{\delta}_i^* = \arg\min_{\|\boldsymbol{\delta}_i\|_p \leq \rho} \mathcal{L}_{\text{CE}}\left(\hat{f}_{\boldsymbol{\theta}}(\boldsymbol{x}_i + \boldsymbol{\delta}_i), y_i'\right), \; y_i' = \arg\max_{k_i \neq y_i} \hat{f}_{k_i}(\boldsymbol{x}_i; \boldsymbol{\theta}), i = 1, \ldots, N, \tag{6}$$

where $\lambda_1, \lambda_2, \lambda_3 > 0$ are hyper-parameters to adjust the importance of each loss function. This bi-level optimization is dubbed Neural Polarizer based backdoor Defense (**NPD**).

**Variants.** To comprehensively evaluate the performance of NPD, we also provide two variants, with the relaxation that if the target label $T$ is known. One is that approximating the trigger by the targeted adversarial perturbation for each benign sample in Eq. (6), *i.e.*, $\Delta \approx \boldsymbol{\delta}_i^* = \arg\min_{\|\boldsymbol{\delta}_i\|_p \leq \rho} \mathcal{L}_{\text{CE}}(f(\boldsymbol{x}_i + \boldsymbol{\delta}_i), T)$, dubbed **NPD-TP**. The other is approximating the trigger by the targeted universal adversarial perturbation (TUAP) [25] for all training samples in fine-tuning, dubbed **NPD-TU**. We empirically found that the targeted adversarial perturbation is a better surrogate to the trigger than the untargeted ones.

**Optimization algorithm.** We proposed Algorithm 1 to solve the above optimization problem. Specifically, NPD solves problem (6) by alternatively updating the surrogate target label $y'$, the perturbation $\boldsymbol{\delta}$ and $\boldsymbol{\theta}$ as follows:

- **Inner minimization:** Given parameter $\boldsymbol{\theta}$ of the neural polarizer, we first estimate the target label for sample $\boldsymbol{x}_i$ by $y_i' = \arg\max_{k_i \neq y_i} \hat{f}_{k_i}(\boldsymbol{x}_i; \boldsymbol{\theta})$. Then, the targeted Project Gradient Descent (PGD) [27] is employed to generate the perturbation $\boldsymbol{\delta}_i^*$ via Eq. (5).

- **Outer minimization:** Given $y'$ and $\boldsymbol{\delta}^*$ for each sample in a batch, the $\boldsymbol{\theta}$ can be updated by taking one stochastic gradient descent [3] (SGD) step *w.r.t.* the outer minimization objective in Eq. (6).

## 4 Experiments

### 4.1 Implementation details

**Attack settings.** We evaluate the proposed method on eight famous backdoor attacks, including BadNets [11] (BadNets-A2O and BadNets-A2A refer to attacking one target label and all labels, respectively), Blended attack (Blended) [5], Input-aware dynamic backdoor attack (Input-aware)[28], Low frequency attack (LF) [46], Sample-specific backdoor attack (SSBA) [20], Trojan backdoor

---

**Algorithm 1** Neural Polarizer based Backdoor Defense (NPD)

---

1: **Input:** Training set $\mathcal{D}_{bn}$, backdoored model $f_{\boldsymbol{w}}$, neural polarizer $g_{\boldsymbol{\theta}}$, learning rate $\eta > 0$, perturbation bound $\rho > 0$, norm $p$, hyper-parameters $\lambda_1, \lambda_2, \lambda_3 > 0$, warm-up epochs $\mathcal{T}_0$, training epochs $\mathcal{T}$, number of PGD steps $T_{adv}$.

2: **Output:** Model $\hat{f}(\boldsymbol{w}, \boldsymbol{\theta})$.

3: Initialize $\boldsymbol{\theta}$ to be an identity function, fix $\boldsymbol{w}$, and construct the composed network $\hat{f}(\boldsymbol{w}, \boldsymbol{\theta})$.

4: Warm-up: Train $\hat{f}(\boldsymbol{w}, \boldsymbol{\theta})$ with $\mathcal{L}_{\mathrm{CE}}(D_{bn})$ for $\mathcal{T}_0$ epochs.

5: **for** $t = 0, ..., \mathcal{T} - 1$ **do**

6:     **for** mini-batch $\mathcal{B} = \{(\boldsymbol{x}_i, y_i)\}_{i=1}^{b} \subset \mathcal{D}_{bn}$ **do**

7:         Compute $\{y_i'\}_{i=1}^{b}$ by Eq. (4);

8:         Generate perturbations $\{(\boldsymbol{\delta_i})\}_{i=1}^{b}$ with $\|\boldsymbol{\delta_i}\|_{\boldsymbol{p}} \leq \rho$ and $\{y_i'\}_{i=1}^{b}$ by targeted PGD attack [27] via Eq. (5);

9:         Update $\boldsymbol{\theta}$ via outer minimization of Eq. (6) by SGD.

10:     **end for**

11: **end for**

12: **return** Model $\hat{f}(\boldsymbol{w}, \boldsymbol{\theta})$.

---

attack (Trojan) [24], and Warping-based poisoned networks (WaNet) [29]. We follow the default attack configuration as in BackdoorBench [42] for a fair comparison. The poisoning ratio is set to 10% in comparison with SOTA defenses. These attacks are conducted on three benchmark datasets: CIFAR-10 [16], Tiny ImageNet [18], and GTSRB [34]. We test all attacks on PreAct-ResNet18 [13] and VGG19-BN [33].

**Defense settings.** We compare the proposed methods with six SOTA backdoor defense methods, *i.e.*, Fine-pruning (FP) [23], NAD [19], NC [35], ANP [43], i-BAU [45], and EP [48]. All these defenses have access to 5% benign training samples. The training hyperparameters are adjusted based on BackdoorBench [42]. We evaluate the proposed method under two proposed scenarios and compare our NPD-TU, NPD-TP, and NPD with SOTA defenses. For the ablation study, we focus solely on NPD, which represents a more generalized scenario. We apply an $l_2$ norm constraint to the adversarial perturbations, with a perturbation bound of 3 for CIFAR-10 and GTSRB datasets, and 6 for Tiny ImageNet. We train the neural polarizer for 50 epochs with batch size 128 and learning rate 0.01 on each dataset and the transformation block is inserted before the third convolution layer of the fourth layer for PreAct-ResNet18. The loss hyper-parameters $\lambda_1, \lambda_2, \lambda_3$ are set to $1, 0.4, 0.4$ for NPD, and $1, 0.1, 0.1$ for NPD-TU and NPD-TP. More implementation details on SOTA attacks, defenses, and our methods can be found in Section B of **Appendix**.

**Evaluation metric.** In this work, we use clean ACCuracy (ACC), Attack Success Rate (ASR), and Defense Effectiveness Rating (DER) as evaluation metrics to assess the performance of different defenses. ACC represents the accuracy of clean samples while ASR measures the ratio of successfully misclassified backdoor samples to the target label. Defense Effectiveness Rating (DER $\in [0,1]$ [49]) is a comprehensive measure that considers both ACC and ASR:

$$\mathrm{DER} = [\max(0, \Delta\mathrm{ASR}) - \max(0, \Delta\mathrm{ACC}) + 1]/2, \tag{7}$$

where $\Delta\mathrm{ASR}$ denotes the decrease of ASR after applying defense, and $\Delta\mathrm{ACC}$ denotes the drop in ACC following defense. Higher ACC, lower ASR, and higher DER indicate better defense performance. Note that in comparison with SOTA defenses, the one achieving the best performance is highlighted in **boldface**, while the second-best result is indicated by underlining. We provide PyTorch and MindSpore implementations of NPD.

### 4.2 Main results

Table 1 and Table 2 showcase the defense performance of the proposed method in comparison to six SOTA defense methods on CIFAR-10 and Tiny ImageNet. The following observations can be made:

- **Our methods show superior performance in terms of DER for almost all attacks compared to SOTA defenses.** Conversely, FP and ANP excel in maintaining high ACC, but they struggle to eliminate backdoors in strong attacks like Blended and LF. NC's emphasis on minimal universal

Table 1: Comparison with the state-of-the-art defenses on CIFAR-10 dataset with 5% benign data and 10% poison ratio on PreAct-ResNet18 (%).

| ATTACK | Backdoored | | | FP [23] | | | NAD [19] | | | NC [35] | | | ANP [43] | | |
|---|---|---|---|---|---|---|---|---|---|---|---|---|---|---|---|
| | ACC | ASR | DER | ACC | ASR | DER | ACC | ASR | DER | ACC | ASR | DER | ACC | ASR | DER |
| BadNets-A2O [11] | 91.82 | 93.79 | N/A | **91.77** | 0.84 | **96.45** | 88.82 | 1.96 | 94.42 | 90.27 | 1.62 | 95.31 | 91.65 | 3.83 | 94.89 |
| BadNets-A2A [11] | 91.89 | 74.42 | N/A | 92.05 | 1.31 | 86.56 | 90.73 | 1.61 | 85.82 | 89.79 | 1.11 | 85.60 | 92.33 | 2.56 | 85.93 |
| Blended [5] | 93.69 | 99.76 | N/A | 92.74 | 10.17 | 94.32 | 92.25 | 47.64 | 75.34 | 93.69 | 99.76 | 50.00 | 93.45 | 47.14 | 76.19 |
| Input-Aware [28] | 94.03 | 98.35 | N/A | 94.05 | 1.62 | 98.36 | 94.08 | 0.92 | 98.71 | 93.84 | 10.48 | 93.84 | 94.06 | 1.57 | 98.39 |
| LF [46] | 93.01 | 99.06 | N/A | 92.05 | 21.32 | 88.39 | 91.72 | 75.47 | 61.15 | 93.01 | 99.06 | 50.00 | 92.53 | 26.38 | 86.10 |
| SSBA [20] | 92.88 | 97.07 | N/A | 92.21 | 20.27 | 88.06 | 92.15 | 70.77 | 62.78 | 92.88 | 97.07 | 50.00 | 92.02 | 16.18 | 90.01 |
| Trojan [24] | 93.47 | 99.99 | N/A | 92.24 | 67.73 | 65.51 | 92.25 | 5.77 | 96.47 | 91.85 | 51.03 | 73.67 | 92.71 | 84.82 | 57.20 |
| WaNet [29] | 92.80 | 98.90 | N/A | 92.94 | **0.66** | **99.12** | 93.07 | 0.73 | 99.08 | 92.80 | 98.90 | 50.00 | 93.24 | 1.54 | 98.68 |
| Avg | 92.95 | 95.17 | N/A | 92.51 | 15.49 | 89.62 | 91.88 | 25.61 | 84.24 | 92.27 | 57.38 | 68.55 | **92.75** | 23.00 | 85.98 |

| ATTACK | i-BAU [45] | | | EP [48] | | | NPD-TU(**Ours**) | | | NPD-TP(**Ours**) | | | NPD(**Ours**) | | |
|---|---|---|---|---|---|---|---|---|---|---|---|---|---|---|---|
| | ACC | ASR | DER | ACC | ASR | DER | ACC | ASR | DER | ACC | ASR | DER | ACC | ASR | DER |
| BadNets-A2O [11] | 87.43 | 4.48 | 92.46 | 89.80 | 1.26 | 95.26 | 90.81 | 1.44 | 95.67 | 90.90 | **0.62** | 96.12 | 88.93 | 1.26 | 94.82 |
| BadNets-A2A [11] | 89.39 | 1.29 | 85.32 | 88.72 | 3.00 | 84.12 | 91.66 | 0.82 | 86.68 | 92.54 | **0.04** | 87.19 | 91.41 | 0.89 | 86.52 |
| Blended [5] | 89.43 | 26.82 | 84.34 | 91.94 | 48.22 | 74.89 | 91.88 | 0.03 | 98.96 | 91.33 | 0.83 | 98.28 | 91.18 | 0.41 | 98.42 |
| Input-Aware [28] | 89.91 | 0.02 | 97.10 | 93.68 | 2.88 | 97.56 | 92.01 | 0.14 | 98.09 | 93.24 | 0.00 | 98.78 | 89.57 | 0.11 | 96.89 |
| LF [46] | 88.92 | 11.99 | 91.49 | 91.97 | 84.73 | 56.64 | 91.42 | 0.01 | 98.73 | 91.92 | 0.08 | 98.94 | 90.06 | 0.21 | 97.95 |
| SSBA [20] | 86.53 | 2.89 | 93.91 | 91.67 | 4.33 | 95.76 | 91.61 | 2.46 | 96.67 | 91.82 | 0.83 | 97.59 | 90.88 | 2.77 | 96.15 |
| Trojan [24] | 89.29 | 0.54 | 97.63 | 92.32 | 2.49 | 98.18 | 92.59 | 0.04 | 99.53 | 92.19 | 0.00 | 99.35 | 92.37 | 6.51 | 96.19 |
| WaNet [29] | 90.70 | 0.88 | 97.96 | 90.47 | 96.52 | 50.03 | 92.18 | 3.24 | 97.52 | 92.57 | 7.47 | 95.60 | 91.57 | 0.80 | 98.43 |
| Avg | 88.95 | 6.11 | 92.53 | 91.32 | 30.43 | 81.55 | 91.77 | **1.02** | 96.48 | 92.06 | 1.23 | 96.52 | 90.75 | 1.62 | 95.67 |

Table 2: Comparison with the state-of-the-art defenses on Tiny ImageNet dataset with 5% benign data and 10% poison ratio on PreAct-ResNet18 (%).

| ATTACK | Backdoored | | | FP [23] | | | NAD [19] | | | NC [35] | | | ANP [43] | | |
|---|---|---|---|---|---|---|---|---|---|---|---|---|---|---|---|
| | ACC | ASR | DER | ACC | ASR | DER | ACC | ASR | DER | ACC | ASR | DER | ACC | ASR | DER |
| BadNets-A2O [11] | 56.12 | 99.90 | N/A | 48.81 | 0.66 | 95.96 | 48.35 | 0.27 | 95.93 | **56.12** | 99.90 | 50.00 | 47.34 | **0.00** | 95.56 |
| BadNets-A2A [11] | 55.99 | 27.81 | N/A | 47.88 | 3.19 | 58.26 | 48.29 | 2.30 | 58.91 | 54.12 | 18.72 | 53.61 | 40.70 | 2.39 | 55.07 |
| Blended [5] | 56.49 | 99.67 | N/A | 50.58 | 57.89 | 67.93 | 55.22 | 98.88 | 49.76 | 54.50 | 96.07 | 50.80 | 43.21 | 43.80 | 71.29 |
| Input-Aware [28] | 57.67 | 99.19 | N/A | 52.38 | 0.13 | 96.88 | 57.42 | 0.07 | 99.43 | 53.46 | 2.48 | 96.25 | 50.56 | 0.00 | 96.04 |
| LF [46] | 55.21 | 98.51 | N/A | 48.18 | 63.83 | 63.83 | 49.61 | 58.01 | 67.45 | 53.08 | 90.48 | 52.95 | 41.75 | 65.98 | 59.54 |
| SSBA [20] | 55.97 | 97.69 | N/A | 48.06 | 52.25 | 68.76 | 47.67 | 69.47 | 59.96 | 53.30 | 0.26 | 97.38 | 41.83 | 14.24 | 84.65 |
| Trojan [24] | 56.48 | 99.97 | N/A | 45.96 | 8.88 | 90.28 | 48.83 | 1.01 | 99.65 | 54.43 | 1.54 | 98.19 | 45.36 | 0.53 | 94.16 |
| WaNet [29] | 57.81 | 96.50 | N/A | 50.35 | 1.37 | 93.83 | 50.02 | 0.87 | 93.92 | 57.81 | 96.50 | 50.00 | 30.34 | **0.00** | 84.51 |
| Avg | 56.47 | 89.90 | N/A | 49.02 | 23.53 | 79.47 | 50.68 | 28.86 | 77.63 | 54.60 | 50.74 | 68.65 | 42.64 | 15.87 | 80.10 |

| ATTACK | i-BAU [45] | | | EP [48] | | | NPD-TU(**Ours**) | | | NPD-TP(**Ours**) | | | NPD(**Ours**) | | |
|---|---|---|---|---|---|---|---|---|---|---|---|---|---|---|---|
| | ACC | ASR | DER | ACC | ASR | DER | ACC | ASR | DER | ACC | ASR | DER | ACC | ASR | DER |
| BadNets-A2O [11] | 51.63 | 95.92 | 49.74 | 54.00 | 0.02 | **98.88** | 47.23 | 0.01 | 95.50 | 49.89 | 1.28 | 96.19 | 49.79 | 2.51 | 95.53 |
| BadNets-A2A [11] | 53.52 | 12.89 | 56.22 | 54.79 | 1.28 | 62.67 | 46.81 | 1.96 | 58.34 | 49.79 | 3.31 | 59.15 | 49.94 | 5.57 | 58.10 |
| Blended [5] | 50.76 | 95.58 | 49.18 | 56.32 | 88.88 | 55.31 | 46.24 | 0.00 | 94.71 | 49.72 | 0.18 | 96.36 | 49.62 | 0.12 | 96.34 |
| Input-Aware [28] | 55.49 | 0.46 | 98.27 | 57.33 | 0.03 | 99.41 | 49.54 | 0.27 | 95.39 | 53.88 | 0.04 | 97.68 | 53.75 | 5.93 | 94.67 |
| LF [46] | 53.65 | 94.27 | 51.34 | 54.86 | 93.20 | 52.48 | 46.04 | 0.00 | 94.67 | 49.20 | 0.30 | 96.10 | 49.94 | 2.48 | 95.38 |
| SSBA [20] | 52.39 | 84.64 | 54.73 | 55.56 | 66.67 | 65.30 | 46.56 | 0.00 | 94.14 | 49.04 | 0.00 | 95.38 | 49.25 | 0.01 | 95.48 |
| Trojan [24] | 51.85 | 99.15 | 48.09 | 54.47 | 0.12 | **98.92** | 48.56 | 0.00 | 96.02 | 49.61 | 0.05 | 96.52 | 49.43 | 0.51 | 96.20 |
| WaNet [29] | 53.04 | 69.82 | 60.95 | 57.06 | 0.20 | **97.77** | 48.52 | 0.01 | 93.60 | 51.88 | 0.82 | 94.88 | 52.64 | 0.24 | 95.54 |
| Avg | 52.79 | 69.09 | 58.57 | **55.55** | 31.30 | 78.84 | 47.44 | **0.28** | 90.30 | 50.38 | 0.75 | **91.53** | 50.54 | 2.17 | 90.90 |

adversarial perturbation renders it ineffective against sample-specific attacks and those utilizing large norm triggers. I-BAU shows similar performance in removing backdoors with an average DER of 92.53%, but it leads to a significant decrease in ACC, likely due to training the entire network by adversarial training.

- **Defense performance of NPD-TU and NPD-TP are better than NPD.** When the target label is known, the model only needs to find perturbations for that specific label, simplifying trigger identification and unlearning. These two methods outperform NPD, except for WaNet, which is a transformation-based attack without visible triggers. Fully perturbing the network proves more effective than solely unlearning targeted triggers in WaNet.

- **NPD-TU is effective for trigger-additive attacks while NPD-TP is expert in defending against sample-specific attacks.** It can be observed by comparing defense results on different attacks like Blended and SSBA on CIFAR-10. This demonstrates that the applicability of different strategies varies across different attack scenarios.

- **Defense performance is robust across all attacks on Tiny ImageNet.** Similar to the results on CIFAR-10, our method outperforms other methods in terms of ASR and DER for all backdoor

Table 3: Defense performance under different components of losses.

| $\mathcal{L}_{bce1}$ | $\mathcal{L}_{bce2}$ | $\mathcal{L}_{asr}$ | BadNets-A2O [11] ACC | ASR | Blended [5] ACC | ASR | LF [46] ACC | ASR |
|---|---|---|---|---|---|---|---|---|
| | | | 91.45 | 1.18 | 92.47 | 99.63 | 92.00 | 95.90 |
| ✓ | | | 90.17 | 1.19 | 91.51 | 2.01 | 90.91 | 9.60 |
| | ✓ | | 90.46 | 0.38 | 91.68 | 18.28 | 91.19 | 1.06 |
| | | ✓ | 90.02 | 0.27 | 91.31 | 98.32 | 91.07 | 0.80 |
| ✓ | ✓ | | 89.56 | 0.21 | 91.09 | 1.73 | 90.47 | 7.63 |
| ✓ | ✓ | ✓ | 88.93 | 1.26 | 91.18 | 0.41 | 90.06 | 0.21 |

Table 4: Defense results in comparison with NPD-UU and NPD-UP.

| ATTACK ↓ | No defense ACC | ASR | NPD-UU ACC | ASR | NPD-UP ACC | ASR | NPD (**Ours**) ACC | ASR |
|---|---|---|---|---|---|---|---|---|
| BadNets-A2O [11] | 91.82 | 93.79 | 79.35 | **0.10** | **90.61** | 1.74 | 88.93 | 1.26 |
| Blended [5] | 93.44 | 97.71 | 86.35 | 10.77 | **92.35** | 3.86 | 91.18 | **0.41** |
| LF [46] | 93.01 | 99.06 | 82.95 | 75.42 | **91.53** | 17.24 | 90.06 | **0.21** |
| SSBA [20] | 92.88 | 97.07 | 84.31 | 52.36 | **91.49** | 14.22 | 90.88 | **2.77** |
| Trojan [24] | 93.47 | 99.99 | 89.81 | 38.11 | **92.61** | 11.43 | 92.37 | **6.51** |
| WaNet [29] | 92.80 | 98.90 | 84.70 | 5.98 | **92.11** | 1.41 | 91.57 | **0.80** |

attacks. Despite a slight decrease in ACC, NPD-TU achieves a remarkably good performance with ASR < 1.5% on average. NPD-TP and NPD perform best in removing backdoors while maintaining model utility.

In summary, our method outperforms other state-of-the-art approaches, showcasing the broad applicability of our proposed method across diverse datasets. Due to space limits, defending results on GTSRB dataset and VGG19-BN network can be found in Section C of **Appendix**.

### 4.3 Analysis

**Effectiveness of each loss term.** We conduct an ablation study to evaluate the contribution of each component of the loss function towards the overall performance on CIFAR-10 dataset. We separately investigate the first and second terms of loss $\mathcal{L}_{bce}$ (see Eq. (3)), denoting them as $\mathcal{L}_{bce1}$ and $\mathcal{L}_{bce2}$, respectively. Throughout the study, we keep the loss $\mathcal{L}_{bn}$ consistent across all experiments, and the result is shown in Table 3. Notably, the loss $l_{bce1}$ plays a significant role in improving the overall performance while removing each component leads to a significant drop in defense in certain cases. This ablation study underscores the importance of each loss component in effectively mitigating different types of attacks.

**Effectiveness of the targeted adversarial perturbations in NPD.** To show the efficacy of the targeted adversarial perturbations in NPD (see Eq. (6)), we compare NPD with its two variants using two types of untargeted perturbation. We refer to adversarial perturbations generated by UAP and standard PGD without a targeted label as NPD-UU (untargeted universal adversarial perturbation) and NPD-UP (untargeted PGD), respectively. As shown in Table 4, NPD-UU and NPD-UP fail to remove backdoors in certain cases although NPD-UP obtains a higher ACC. This result shows the superiority of NPD in removing backdoors.

**Performance of choosing different layers to insert the transformation layer.** We evaluate the influence of choosing different layers to insert the transformation layer by inserting it after each convolution layer of PreAct-ResNet18 network on CIFAR-10 dataset. Figure 3 shows the defense performance under three attacks. The result shows that inserting the transformation layer into the shallower layers results in a decrease in accuracy. This is because even a slight perturbation in the shallow layers can cause significant instability in the final output. However, as the layer goes deeper, the features become more separable, resulting in better defense performance.

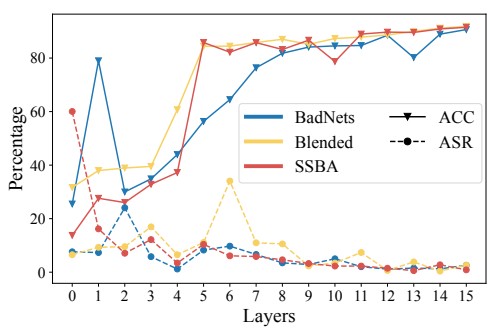

Figure 3: Defense performance of inserting linear transformation into different layers.

**Defense effectiveness under different poisoning ratios.** To investigate the impact of poisoning ratios on defense performance, we conducted experiments on NPD with different poisoning ratios on the CIFAR-10 dataset. As presented in Table 5, there is a slight decrease in ACC as the poisoning ratio increases. Moreover, our approach exhibits a notably stable defense performance across a range of poisoning ratios.

Table 5: Defense results under different poisoning ratio on CIFAR-10 and PreAct-ResNet18(%).

| Poisoning Ratio → ATTACK ↓ | | 5% | | 10% | | 20% | | 30% | | 40% | |
|---|---|---|---|---|---|---|---|---|---|---|---|
| | | No Defense | Ours | No Defense | Ours | No Defense | Ours | No Defense | Ours | No Defense | Ours |
| BadNets-A2O [11] | ACC | 92.35 | 87.99 | 91.82 | 88.93 | 90.17 | 85.77 | 88.32 | 86.76 | 86.16 | 82.31 |
| | ASR | 89.52 | 0.89 | 93.79 | 1.26 | 96.12 | 0.39 | 97.33 | 0.69 | 97.78 | 3.88 |
| Blended [5] | ACC | 93.76 | 91.48 | 93.44 | 91.18 | 93.00 | 91.27 | 92.78 | 90.10 | 91.64 | 88.85 |
| | ASR | 99.31 | 11.06 | 97.71 | 0.41 | 99.92 | 4.41 | 99.98 | 3.78 | 99.96 | 24.92 |
| Input-Aware [28] | ACC | 90.92 | 90.31 | 94.03 | 89.57 | 89.18 | 88.60 | 89.63 | 89.41 | 90.12 | 88.57 |
| | ASR | 94.19 | 0.30 | 98.35 | 0.11 | 97.66 | 3.51 | 97.62 | 3.43 | 98.57 | 0.98 |
| WaNet [29] | ACC | 93.38 | 91.38 | 92.80 | 91.57 | 91.02 | 89.77 | 92.35 | 89.53 | 92.21 | 89.53 |
| | ASR | 97.27 | 0.14 | 98.90 | 0.80 | 94.93 | 0.22 | 99.06 | 1.71 | 99.49 | 2.19 |

Table 6: Results with different number of clean data on CIFAR-10 (%).

| ATTACK | # Clean data → Defense ↓ | 500 | | | 250 | | | 50 | | |
|---|---|---|---|---|---|---|---|---|---|---|
| | | ACC | ASR | DER | ACC | ASR | DER | ACC | ASR | DER |
| BadNets-A2O [11] | i-BAU [45] | 65.41 | 7.74 | 79.82 | 66.05 | 2.02 | 81.03 | 64.60 | 22.10 | 70.27 |
| | ANP [43] | 91.53 | 5.50 | 94.00 | 90.81 | 2.03 | **93.40** | 85.57 | 1.52 | 91.04 |
| | Ours | 87.58 | 1.38 | **94.09** | 87.80 | 0.27 | 92.78 | 86.91 | 0.20 | **92.37** |
| SSBA [20] | i-BAU [45] | 84.21 | 17.73 | 85.33 | 76.84 | 51.90 | 60.38 | 67.48 | 98.37 | 35.21 |
| | ANP [43] | 92.06 | 28.67 | 83.79 | 92.13 | 27.61 | 80.17 | 88.31 | 22.28 | 80.92 |
| | Ours | 90.48 | 0.43 | **97.12** | 90.80 | 10.03 | **88.29** | 89.48 | 8.97 | **88.16** |
| LF [46] | i-BAU [45] | 83.91 | 19.34 | 85.31 | 70.12 | 99.42 | 35.53 | 70.26 | 99.31 | 35.60 |
| | ANP [43] | 92.74 | 46.70 | 76.04 | 92.28 | 18.02 | 84.11 | 89.99 | 18.77 | 82.59 |
| | Ours | 90.21 | 0.99 | **97.63** | 89.28 | 0.50 | **91.37** | 88.18 | 10.83 | **85.65** |

**Defense effectiveness under different clean ratios.** We investigate the sensitivity of clean data on defense performance and compare our NPD with SOTA defenses. As shown in Table 6, NPD is less sensitive to the size of clean data among all the attacks and defenses. Even with only 50 samples, it still maintains acceptable performance. This result shows that our method exhibits minimal reliance on the number of training samples.

**Defense against adaptive attacks.** We evaluate the effectiveness of NPD against adaptive attacks. Consider the attacker knows the defense strategy and the attacker trains the backdoored model with adversarial training (AT). The defense performance against the AT backdoored model under different attacks is shown in Table 7. NPD still performs well on AT model, while i-BAU fails in these attacks. One possible reason is that i-BAU is essentially adversarial training, while NPD adopts dynamic targeted adversarial perturbation, which is different from AT. Compared to the defense against backdoored models with standard training (Table 1), there is a slight ASR increase (from 2.77% to 5.22% for SSBA, for example).

**Defense against all to all attacks.** To demonstrate the effectiveness of our dynamic and sample-specific target label prediction strategy, we conducted experiments involving all-to-all attacks for various backdoor attack methods. In these experiments, each class is treated as the target class. The result is presented in Table 8. As illustrated in the table, our NPD approach successfully eliminates all backdoors, proving its efficacy in multi-label cases. In contrast, i-BAU method fails in most cases.

**Running time comparison.** We measure the runtime of the defense methods on 2500 CIFAR-10 images with batch size 256 and PreAct-ResNet18. The experiments were conducted on one RTX 4090Ti GPU and the results are presented in Table 9. Among these methods, our proposed PND-UN achieves the fastest performance on CIFAR-10, requiring only 56 seconds. It should be noted that our method was trained for 50 epochs, while i-BAU was only trained for 5 epochs.

Table 7: Defense performance against adaptive attacks on CIFAR-10 dataset (%).

Table 8: Defense performance against all2all attacks on CIFAR-10 dataset (%).

| ATTACK | Backdoored | | i-BAU [45] | | NPD | | ATTACK | Backdoored | | i-BAU [45] | | NPD | |
|---|---|---|---|---|---|---|---|---|---|---|---|---|---|
| | ACC | ASR | ACC | ASR | ACC | ASR | | ACC | ASR | ACC | ASR | ACC | ASR |
| Blended [5] | 86.00 | 99.63 | 83.98 | 30.43 | 83.64 | 3.92 | Blended [5] | 91.59 | 83.50 | 88.80 | 15.74 | 91.07 | 6.24 |
| Input-Aware [28] | 84.98 | 94.99 | 83.17 | 71.12 | 83.13 | 4.47 | Input-Aware [28] | 86.60 | 78.55 | 89.50 | 5.37 | 90.10 | 2.10 |
| LF [46] | 84.15 | 94.30 | 84.15 | 94.30 | 83.39 | 4.89 | LF [46] | 91.91 | 84.80 | 88.92 | 33.33 | 90.53 | 4.88 |
| SSBA [20] | 84.34 | 93.32 | 82.74 | 24.54 | 83.38 | 5.22 | SSBA [20] | 91.30 | 85.04 | 91.16 | 26.00 | 91.14 | 1.29 |

Table 9: Running time of different defense methods with 2500 CIFAR-10 images on PreActResNet18.

| Defense (sec.) | FP [23] | NAD [19] | NC [35] | ANP [43] | i-BAU [45] | EP [48] | NPD (**Ours**) |
|---|---|---|---|---|---|---|---|
| CIFAR-10 | 1169.01 | 74.39 | 896.45 | 58.75 | 57.23 | 131.84 | 55.16 |
| Tiny ImageNet | 3357 | 351 | 42512 | 1692 | 887 | 302 | 332 |

## 5 Conclusion

Inspired by the mechanism of optical polarizer, this work proposed a novel backdoor defense method by inserting a learnable neural polarizer as an intermediate layer of the backdoored model. We instantiated the neural polarizer as a lightweight linear transformation and it could be efficiently and effectively learned with limited clean samples to mitigate backdoor effect. To learn a desired neural polarizer, a bi-level optimization problem is proposed by filtering trigger features of poisoned samples while maintaining benign features of both poisoned and benign samples. Extensive experiments demonstrate the effectiveness of our method across all evaluated backdoor attacks and all other defense methods under various datasets and network architectures.

**Limitations and future work.** Although only limited clean data is needed for our method to achieve a remarkable defense performance, the accessibility of clean data is still an important limitation of the proposed method, which may restrict the application of our method. Therefore, a promising direction for future work is to further reduce the requirement of clean data by exploring data-free neural polarizer or learning neural polarizer based on poisoned training data.

**Broader impacts.** Backdoor attacks pose significant threats to the deployment of deep neural networks obtained from untrustworthy sources. This work has made a valuable contribution to the community with an efficient and effective backdoor defense strategy to ease the threat of existing backdoor attacks, even with a very limited set of clean samples, which ensures its practicality. Besides, the innovative defense strategy of learning additional lightweight layers, rather than adjusting the whole backdoored model, may inspire more researchers to develop more efficient and practical defense methods.

**Structure of Appendix.** Detailed algorithms of PGD and UAP, proposed NPD-TU and NPD-TP, along with a theoretical analysis in the kernel space are provided in Section A. Implementation details of experiments are introduced in Section B. More defense results in different datasets and networks are provided in Section C. Defense performance of different structures of neural polarizer is discussed in Section D. Additional experimental results and visualization are presented in Section E.

## 6 Acknowledgments

This work is supported by the National Natural Science Foundation of China under grant No. 62076213, Shenzhen Science and Technology Program under grant No. RCYX20210609103057050, No. ZDSYS20211021111415025, No. GXWD20201231105722002-20200901175001001, Shenzhen Science and Technology Program under grant No. JCYJ20210324120011032, CAAI-Huawei MindSpore Open Fund, and the Guangdong Provincial Key Laboratory of Big Data Computing, the Chinese University of Hong Kong, Shenzhen.

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

# Appendix

This appendix is organized as follows.

- In Section A, we first provide detailed algorithms of the proposed NPD-TU and NPD-TP, along with an introduction to the project gradient descent (PGD) method and UAP method. Then we present an example of backdoor attacks in the kernel space and provide a theoretical analysis of the existence of linear transformation to mitigate backdoors.

- In Section B, we introduce the implementation details, including dataset introduction, implementation details of compared attacks, defenses, and our proposed methods.

- In Section C, we display the defense results in comparison with state-of-the-art (SOTA) defenses on GTSRB dataset, as well as those on CIFAR-10 with the VGG19-BN network.

- In Section D, we discuss different structures of neural polarizers on the defense performance.

- In Section E, we present additional experimental results, including defense experiments conducted on the CIFAR-100 dataset, defense performance across various model architectures, and visualization experiments.

## A  Theoretical analysis and algorithm

### A.1  Introduction to PGD algorithm and UAP

In this section, we provide an introduction to PGD attack [27] and universal adversarial perturbation (UAP) attack [25].

**PGD.**  Projected Gradient Descent (PGD) is a multi-step variant of Fast Gradient Sign Method (FGSM [17], which is projected gradient descent on the negative loss function, *i.e.*, given an input $(\boldsymbol{x}, y)$ and a model $\hat{f}_{\boldsymbol{\theta}}$, the adversarial example is calculated as follows,

$$\boldsymbol{x}^{n+1} = \Pi_{\boldsymbol{x}+\mathcal{S}} \left( \boldsymbol{x}^n + \alpha \operatorname{sgn} \left( \nabla_{\boldsymbol{x}} L_{\mathrm{CE}}(\hat{f}_{\boldsymbol{\theta}}(\boldsymbol{x}), y) \right) \right), \tag{7}$$

where $\Pi_{x+\mathcal{S}}$ denotes the projection on to the $\rho$-ball with $L_p$ norm: $\|\boldsymbol{x}^{n+1} - \boldsymbol{x}\|_p \leq \rho$, and $\boldsymbol{x}^0 = \boldsymbol{x}$, $n = 0, \cdots, N-1$. $\alpha$ is the step size. However, in Section 3 of the main script, we use a confidence-guided targeted PGD approach to obtain the perturbation. Thus, the adversarial example is updated as follows,

$$\boldsymbol{x}^{n+1} = \Pi_{\boldsymbol{x}+\mathcal{S}} \left( \boldsymbol{x}^n - \alpha \operatorname{sgn} \left( \nabla_{\boldsymbol{x}} L_{\mathrm{CE}}(\hat{f}_{\boldsymbol{\theta}}(\boldsymbol{x}), T) \right) \right), \tag{8}$$

where $T$ is the target label for $\boldsymbol{x}$. We denote the adversarial perturbation as follows for convenience:

$$\boldsymbol{\delta} = \boldsymbol{x}^{n+1} - \boldsymbol{x}^n. \tag{9}$$

**UAP.**  The main difference between UAP and PGD is that UAP computes a universal adversarial perturbation for the whole training samples; thus the adversarial example for targeted UAP is updated as follows,

$$\boldsymbol{x}_i^{n+1} = \Pi_{\boldsymbol{x}_i+\mathcal{S}} \left( \boldsymbol{x}_i^n - \alpha \operatorname{sgn} \left( \sum_{i=0}^{b} \nabla_{\boldsymbol{x}_i} L_{\mathrm{CE}}(\hat{f}_{\boldsymbol{\theta}}(\boldsymbol{x}_i), T) \right) \right), \tag{10}$$

where $T$ is the target label for $\boldsymbol{x}$, and $b$ is the number of samples in one training batch. Eq (10) means all the samples in one batch share the sample gradient to update the perturbation.

### A.2  Detailed algorithms of NPD-TU and NPD-TP

In the main script, we introduce the algorithm for NPD. Here, we provide the detailed algorithms for NPD-TU and NPD-TP as in Alg 2 and 3.

---

**Algorithm 2** **N**eural **P**olarizer based Backdoor **D**efense of **T**argeted **U**AP (NPD-TU)

---

1: **Input:** Training set $\mathcal{D}_{bn}$, backdoored model $f_{\boldsymbol{w}}$, neural polarizer $g_{\boldsymbol{\theta}}$, target label $T$, inner steps $N$, learning rate $\eta > 0$, perturbation bound $\rho > 0$, norm $p$, hyper-parameters $\lambda_1, \lambda_2, \lambda_3 > 0$, warm-up epochs $\mathcal{T}_0$, training epochs $\mathcal{T}$.
2: **Output:** Model $\hat{f}(\boldsymbol{w}, \boldsymbol{\theta})$.
3: Initialize $\boldsymbol{\theta}$ to be an identity function, fix $\boldsymbol{w}$, and construct the composed network $\hat{f}(\boldsymbol{w}, \boldsymbol{\theta})$.
4: Warm-up: Train $\hat{f}(\boldsymbol{w}, \boldsymbol{\theta})$ with $\mathcal{L}_{\text{CE}}(D_{bn})$ for $\mathcal{T}_0$ epochs.
5: **for** $t = 0, ..., \mathcal{T} - 1$ **do**
6:     **for** mini-batch $\mathcal{B} = \{(\boldsymbol{x}_i, y_i)\}_{i=1}^b \subset \mathcal{D}_{bn}$ **do**
7:         **for** $n = 0, ..., N - 1$ **do**
8:             Compute adversarial perturbation $\boldsymbol{\delta}$ with $\|\boldsymbol{\delta}\|_{\boldsymbol{p}} \leq \rho$ and target label $T$ by targeted UAP attack [25] via Eq. (10);
9:         **end for**
10:         Update $\boldsymbol{\theta}$ via outer minimization of Eq. (5) in the main script by SGD.
11:     **end for**
12: **end for**
13: **return** Model $\hat{f}(\boldsymbol{w}, \boldsymbol{\theta})$.

---

---

**Algorithm 3** **N**eural **P**olarizer based Backdoor **D**efense of **T**argeted **P**GD (NPD-TP)

---

1: **Input:** Training set $\mathcal{D}_{bn}$, backdoored model $f_{\boldsymbol{w}}$, neural polarizer $g_{\boldsymbol{\theta}}$, target label $T$, inner steps $N$, learning rate $\eta > 0$, perturbation bound $\rho > 0$, norm $p$, hyper-parameters $\lambda_1, \lambda_2, \lambda_3 > 0$, warm-up epochs $\mathcal{T}_0$, training epochs $\mathcal{T}$.
2: **Output:** Model $\hat{f}(\boldsymbol{w}, \boldsymbol{\theta})$.
3: Initialize $\boldsymbol{\theta}$ to be an identity function, fix $\boldsymbol{w}$, and construct the composed network $\hat{f}(\boldsymbol{w}, \boldsymbol{\theta})$.
4: Warm-up: Train $\hat{f}(\boldsymbol{w}, \boldsymbol{\theta})$ with $\mathcal{L}_{\text{CE}}(D_{bn})$ for $\mathcal{T}_0$ epochs.
5: **for** $t = 0, ..., \mathcal{T} - 1$ **do**
6:     **for** mini-batch $\mathcal{B} = \{(\boldsymbol{x}_i, y_i)\}_{i=1}^b \subset \mathcal{D}_{bn}$ **do**
7:         **for** $n = 0, ..., N - 1$ **do**
8:             Compute adversarial perturbations $\{(\boldsymbol{\delta}_i)\}_{i=1}^b$ with $\|\boldsymbol{\delta}_i\|_{\boldsymbol{p}} \leq \rho$ and target label $T$ by targeted UAP attack [25] via Eq. (8);
9:         **end for**
10:         Update $\boldsymbol{\theta}$ via outer minimization of Eq. (5) in the main script by SGD.
11:     **end for**
12: **end for**
13: **return** Model $\hat{f}(\boldsymbol{w}, \boldsymbol{\theta})$.

---

### A.3 Theoretical analysis under kernel space

In this section, we demonstrate the existence of linear transformation in reproducing kernel Hilbert space (RKHS), a very successful tool in various areas of machine learning. For simplicity, we consider a binary classifier $h(\cdot, m) : \mathcal{X} \times \{0, 1\} \to \mathbb{R}$ in a reproducing kernel Hilbert space $\mathcal{H}_{XM}$ generated by kernel $\kappa_{XM}$, where $m$ is a binary poison indicator, *i.e.*, $m = 1$ means adding trigger to the input and $m = 0$ means not adding trigger. Denote the feature of $(\boldsymbol{x}, m)$ in kernel space by $\phi_{XM}(\boldsymbol{x}, m)$. Given a dataset $\mathcal{D}$ and a distribution of poison indicator $\mathcal{M}$, the poisoned classifier is optimized by minimizing a mean squared error loss:

$$h_{bd} = \underset{h \in \mathcal{H}_{XM}}{\arg\min} \mathbb{E}_{\boldsymbol{x} \sim \mathcal{D}, m \sim \mathcal{M}} (h(\boldsymbol{x}, m) - y)^2. \tag{11}$$

Then, by the theorem in [38] which shows that there exists a subspace of $\mathcal{H}_{XM}$ in which a functin $h$ satisfies $\text{Cov}(h(\boldsymbol{x}, m), m) = 0$, we provide the following lemma:

**Lemma 1.** *Assume that $\mathbb{P}(m = 0) \in (0, 1)$ and $\phi_{XM}(\boldsymbol{x}_i, m) \neq \phi_{XM}(\boldsymbol{x}_j, m)$ if $\boldsymbol{x}_i \neq \boldsymbol{x}_j$. Given a poisoned model $h_{bd}$ trained by minimizing (11), there exists a non-trivial linear projection operator $P$ such that*

$$Cov(\langle \hat{\phi}_{XM}(\boldsymbol{x}, m), h_{bd} \rangle_{\mathcal{H}_{XM}}, m) = 0,$$

*where $\hat{\phi}_{XM}(\boldsymbol{x}, m) = P\phi_{XM}(\boldsymbol{x}, m)$ is the projected feature of $\phi_{XM}(\boldsymbol{x}, m)$ and $\langle \cdot, \cdot \rangle_{XM}$ is the inner product in $\mathcal{H}_{XM}$.*

*Proof.* To prove Lemma 1, we first recall that [38] says that the projection operator can be constructed by the eigenfunctions of the operator $\Sigma_{(XM)M}\Sigma_{M(XM)}$ where $\Sigma_{(XM)M}$ is the Covariance operator between $\phi_{XM}(\boldsymbol{x}, m)$ and $\phi_M(m)$ for $\boldsymbol{x} \in \mathcal{D}$ and $m \in \mathcal{M}$. Then, under the assumption that $\mathbb{P}(m = 0) \in (0, 1)$ and $\phi_{XM}(\boldsymbol{x}_i, m) \neq \phi_{XM}(\boldsymbol{x}_j, m)$ if $\boldsymbol{x}_i \neq \boldsymbol{x}_j$, the Covariance operator $\Sigma_{(XM)M}$ is a non-zero operator by its definition and the projection operator generated by $\Sigma_{(XM)M}$ is non-trivial. $\square$

The first assumption in Lemma 1 requires that the backdoored classifier $h_{bd}$ is trained on a partially poisoned dataset to preserve its accuracy on clean samples. The second assumption in Lemma 1 requires that the feature of a poisoned sample retains some information of the original clean sample and is not completely overwritten by the trigger. Under these conditions, the Lemma 1 states that there exists a non-trivial projection operator such that the $h_{bd}$'s prediction for a transformed feature $P\phi_{XM}(\boldsymbol{x}, m)$ is uncorrelated with poisoning $\boldsymbol{x}$ or not. Therefore, such a linear projection operator breaks the correlation between $m$ and the prediction of $\boldsymbol{x}$ without modifying $h_{bd}$, indicating the existence of a neural polarizer in $\mathcal{H}_{XM}$. Although a complex neural network for multi-class classification problems is usually beyond $\mathcal{H}_{XM}$, Lemma 1 provides insights into the existence of an effective neural polarizer.

# B  More implementation details

In this section, we present the implementation details, which include an introduction to the datasets used, the implementation specifics of the attacks and defenses compared, as well as our proposed methods. All experiments are run on one RTX 3090Ti GPU and are repeated over five runs with different random seeds.

**Datasets.**   We evaluate the effectiveness of our method using three datasets: CIFAR-10 [16], Tiny ImageNet [18], and GTSRB [34], following the benchmarks established in [42].

CIFAR-10 comprises 60,000 images distributed across ten classes. The training set consists of 5,000 images per class, while the test set contains 1,000 images per class. Each image in the CIFAR-10 dataset has dimensions of $32 \times 32$ pixels.

Tiny ImageNet is a subset of the larger ImageNet dataset [7]. It encompasses 100,000 training samples and 10,000 testing samples across 200 classes. The images in Tiny ImageNet have dimensions of $64 \times 64$ pixels.

GTSRB consists of 39,209 training images and 12,630 testing images, categorized into 43 classes. The image size for each sample in the GTSRB dataset is $32 \times 32$ pixels.

**Models.**   We evaluate the performance of our method in conjunction with two variations on the PreActResNet18 [13] and VGG19-BN [33] networks. Furthermore, we compare our method with SOTA defense methods on three datasets and the two networks, utilizing a $10\%$ poisoning ratio and a $5\%$ ratio of clean samples for defense.

**Attack details.**   We introduce more details about the backdoor attack implementation here. We follow BackdoorBench to categorize the eight backdoor attacks according to various criteria as in Table 10.

Table 10: Categories of the eight backdoor attack algorithms.

| ATTACK | Fusion pattern | | Size of trigger | | Visibility of trigger | | Variability of trigger | | Num of target classes | |
|---|---|---|---|---|---|---|---|---|---|---|
| | Additive | Non-Additive | Local | Global | Visible | Invisible | agnostic | specific | All2one | All2all |
| BadNets-A2O [11] | ✓ | | ✓ | | ✓ | | ✓ | | ✓ | |
| BadNets-A2A [11] | ✓ | | ✓ | | ✓ | | ✓ | | | ✓ |
| Blended [5] | | ✓ | | ✓ | ✓ | | ✓ | | ✓ | |
| Input-Aware [28] | ✓ | | ✓ | | ✓ | | | ✓ | ✓ | |
| LF [46] | | ✓ | | ✓ | | ✓ | | ✓ | ✓ | |
| SSBA [20] | | ✓ | | ✓ | | ✓ | | ✓ | ✓ | |
| Trojan [24] | ✓ | | ✓ | | ✓ | | ✓ | | ✓ | |
| WaNet [29] | | ✓ | | ✓ | | ✓ | | ✓ | ✓ | |

For BadNets-A2O and BadNets-A2A [11], we patch a $3 \times 3$ white square in the lower right corner of the images for CIFAR-10 and GTSRB datasets, and $6 \times 6$ white square for Tiny ImageNet dataset. For Blended [5], we blend the poisoned samples with a Hello-Ketty image which has the same size as the poisoned samples, and the blended ratio is set to $0.2$.

**Defense details.** The seven SOTA defense methods can be categorized into two types based on the available resources for the defender. EP [48] is a data-free method that assumes the defender only has access to a backdoored model without any clean data. On the other hand, the remaining methods assume that the defender can obtain a subset of clean samples along with a backdoored model, which aligns with our scenario. In our experiments, a learning rate of 0.01 and a batch size of 256 were used for all SOTA defense methods based on fine-tuning. For ANP [43], the pruning threshold was set to 0.4, as we found that the recommended threshold of 0.2 was inadequate for removing backdoors effectively. Regarding the fine-tuning based SOTA methods, the training epochs were set to 100 for CIFAR-10 and Tiny ImageNet, and 50 for the GTSRB dataset. All other settings remained consistent with those specified in BackdoorBench [42].

In our NPD method, we used a learning rate of 0.01 with a weight decay of 0.0005 and a momentum of 0.9. The training process consisted of 50 epochs for all three datasets, with a warm-up period of 5 epochs. During each epoch, the PGD algorithm was employed to search for adversarial perturbations in each batch. The PGD algorithm utilized a learning rate of 0.1. To constrain the adversarial perturbations on the CIFAR-10 dataset, we applied an $l_2$ norm with a bound of 3.

## C   Defense results in comparison to SOTA defenses on GTSRB and VGG19-BN

To further investigate the generalization performance of our NPD on different datasets and networks, we conducted an analysis on the GTSBR dataset and CIFAR-10 on VGG19-BN network. We compared the performance of NPD with various defense methods against different attacks, and the results are presented in Table 11 and Table 12.

As shown in Table 11, our experimental findings consistently demonstrate the superiority of our proposed methods over state-of-the-art defenses in terms of DER and ASR across almost all attacks, except for BadNet and Trojan. Notably, NAD exhibits the best performance against the Trojan attack, which can be attributed to the strong single trigger pattern adopted by Trojan. NAD effectively reverses the trigger through optimization, making it easier to detect and mitigate the Trojan attack. Furthermore, while NPD-TU and NPD-TP exhibit superiority in terms of a low ASR, NPD demonstrates a comparable trade-off between ACC and ASR. On average, NPD achieves a DER of $96.76\%$, which is $6.26\%$ higher than the best SOTA method, i-BAU. The higher DER achieved by NPD on the remaining attacks further validates the effectiveness of our methods in detecting and mitigating various backdoor attacks compared to existing defenses.

Regarding the performance of the VGG19-BN network, as shown in Table 12, NPD-TU and NPD outperform all other defense methods, underscoring the effectiveness of our proposed defense methods in removing backdoor attacks. However, NPD-TU fails to defend against the Input-Aware attack, which is similar to the performance of i-BAU defense. This highlights the importance of incorporating adversarial attacks into the defense strategy, as UAP does not always guarantee successful defense. Nonetheless, our proposed method exhibits comparable and stable performance across almost all attacks.

In summary, our analysis of the GTSBR dataset and VGG19-BN network showcases the robust generalization performance of NPD against various backdoor attacks. The results underscore the effectiveness of our proposed defense methods, especially NPD-TP and NPD, in removing backdoor attacks.

## D   Different structures of neural polarizers on the defense performance

In this section, we demonstrate the effectiveness of our Conv-BN structure in implementing the neural polarizer for defense against backdoors. We compare it with two Conv-BN structures, a Linear transformation layer, and two denoising layers used in two adversarial defense works: OSAD [31]

Table 11: Comparison with the state-of-the-art defenses on GTSRB dataset with 5% benign data and 10% poison ratio on PreAct-ResNet18 (%).

| ATTACK | Backdoored | | | FP [23] | | | NAD [19] | | | NC [35] | | | ANP [43] | | |
|---|---|---|---|---|---|---|---|---|---|---|---|---|---|---|---|
| | ACC | ASR | DER | ACC | ASR | DER | ACC | ASR | DER | ACC | ASR | DER | ACC | ASR | DER |
| BadNets-A2O [11] | 96.35 | 95.02 | N/A | 98.12 | 0.00 | 97.51 | 97.54 | 79.94 | 57.54 | 93.47 | 0.02 | 96.06 | 96.79 | 0.21 | 97.40 |
| BadNets-A2A [11] | 97.05 | 92.33 | N/A | 98.11 | 0.51 | 95.91 | 97.84 | 2.46 | 94.94 | 94.05 | 0.50 | 94.42 | 96.73 | 50.39 | 70.81 |
| Blended [5] | 97.97 | 99.67 | N/A | 98.20 | 68.45 | 65.61 | 97.98 | 99.29 | 50.19 | 48.34 | 5.70 | 72.17 | 98.25 | 99.94 | 50.00 |
| Input-Aware [28] | 97.17 | 97.09 | N/A | 97.98 | 0.22 | 98.43 | 97.47 | 65.55 | 65.77 | 95.79 | 1.08 | 97.32 | 96.20 | 0.00 | 98.06 |
| LF [46] | 97.97 | 99.58 | N/A | 97.87 | 69.19 | 65.15 | 98.24 | 79.76 | 59.91 | 92.22 | 0.18 | 96.82 | 98.03 | 60.36 | 69.61 |
| SSBA [20] | 98.31 | 99.77 | N/A | 98.47 | 60.19 | 69.79 | 98.37 | 96.95 | 51.41 | 90.75 | 1.51 | 95.35 | 98.36 | 98.98 | 50.40 |
| Trojan [24] | 98.33 | 100.00 | N/A | 98.00 | 42.08 | 78.79 | 98.01 | 0.10 | 99.79 | 92.29 | 0.02 | 96.97 | 98.17 | 86.92 | 56.46 |
| WaNet [29] | 95.71 | 98.20 | N/A | 98.88 | 0.28 | 98.96 | 98.32 | 0.04 | 99.08 | 96.34 | 0.01 | 99.10 | 97.42 | 0.18 | 99.01 |
| Avg | 97.35 | 97.71 | N/A | 98.20 | 30.12 | 83.79 | 97.97 | 53.01 | 72.35 | 87.90 | 1.13 | 93.56 | 97.49 | 49.62 | 74.04 |

| ATTACK | i-BAU [45] | | | EP [48] | | | NPD-TU(Ours) | | | NPD-TP(Ours) | | | NPD(Ours) | | |
|---|---|---|---|---|---|---|---|---|---|---|---|---|---|---|---|
| | ACC | ASR | DER | ACC | ASR | DER | ACC | ASR | DER | ACC | ASR | DER | ACC | ASR | DER |
| BadNets-A2O [11] | 96.35 | 0.00 | 97.51 | 96.53 | 1.38 | 96.82 | 97.01 | 0.01 | 97.51 | 97.78 | 8.19 | 93.42 | 95.89 | 4.62 | 94.97 |
| BadNets-A2A [11] | 95.30 | 0.43 | 95.08 | 96.45 | 1.40 | 95.17 | 96.21 | 0.00 | 95.74 | 97.09 | 0.00 | 95.73 | 97.09 | 9.33 | 91.50 |
| Blended [5] | 95.04 | 96.39 | 50.17 | 95.42 | 100.00 | 48.73 | 97.46 | 0.00 | 99.58 | 97.25 | 0.00 | 99.48 | 97.32 | 2.44 | 98.29 |
| Input-Aware [28] | 96.03 | 0.00 | 97.98 | 92.98 | 0.10 | 96.40 | 96.03 | 6.52 | 94.71 | 97.13 | 0.16 | 98.44 | 95.72 | 1.72 | 96.96 |
| LF [46] | 88.69 | 7.43 | 91.44 | 96.40 | 99.15 | 49.43 | 97.60 | 0.00 | 99.61 | 97.28 | 0.00 | 99.44 | 97.30 | 0.77 | 99.07 |
| SSBA [20] | 87.27 | 0.18 | 94.28 | 97.59 | 99.32 | 49.87 | 97.94 | 0.00 | 99.70 | 97.56 | 0.00 | 99.51 | 98.02 | 3.45 | 98.01 |
| Trojan [24] | 93.66 | 0.00 | 97.66 | 97.46 | 0.05 | 99.54 | 97.47 | 0.00 | 99.57 | 96.37 | 0.08 | 98.98 | 96.47 | 0.00 | 99.07 |
| WaNet [29] | 97.50 | 0.26 | 98.97 | 97.26 | 26.92 | 85.64 | 96.99 | 13.55 | 92.33 | 97.66 | 0.00 | 99.10 | 97.74 | 7.77 | 95.22 |
| Avg | 93.73 | 13.09 | 90.50 | 96.26 | 41.04 | 77.79 | 97.09 | 2.51 | 97.47 | 97.15 | 1.05 | 98.22 | 96.94 | 3.76 | 96.76 |

Table 12: Comparison with the state-of-the-art defenses on CIFAR-10 dataset with 5% benign data and 10% poison ratio on VGG19-BN (%).

| ATTACK | Backdoored | | | FP [23] | | | NAD [19] | | | NC [35] | | | ANP [43] | | |
|---|---|---|---|---|---|---|---|---|---|---|---|---|---|---|---|
| | ACC | ASR | DER | ACC | ASR | DER | ACC | ASR | DER | ACC | ASR | DER | ACC | ASR | DER |
| BadNets-A2O [11] | 90.42 | 94.43 | N/A | 89.11 | 12.39 | 90.36 | 86.80 | 5.77 | 92.52 | 88.97 | 5.63 | 93.68 | 90.44 | 87.64 | 53.40 |
| BadNets-A2A [11] | 91.16 | 84.39 | N/A | 89.70 | 1.91 | 90.51 | 88.15 | 1.60 | 89.89 | 91.16 | 84.39 | 50.00 | 91.29 | 81.87 | 51.26 |
| Blended [5] | 91.60 | 96.68 | N/A | 89.60 | 93.14 | 50.77 | 87.45 | 86.98 | 52.78 | 91.91 | 99.50 | 50.00 | 91.69 | 97.37 | 50.00 |
| Input-Aware [28] | 88.66 | 94.58 | N/A | 91.55 | 14.57 | 90.01 | 91.09 | 14.06 | 90.26 | 89.70 | 97.02 | 50.00 | 89.67 | 36.76 | 78.91 |
| LF [46] | 83.28 | 13.83 | N/A | 88.18 | 1.29 | 56.27 | 85.08 | 3.07 | 55.38 | 88.33 | 1.22 | 56.31 | 89.20 | 1.34 | 56.24 |
| SSBA [20] | 90.85 | 95.11 | N/A | 89.26 | 65.33 | 64.09 | 88.11 | 52.22 | 70.08 | 90.85 | 95.11 | 50.00 | 91.11 | 76.00 | 59.56 |
| Trojan [24] | 91.57 | 100.00 | N/A | 90.04 | 29.71 | 84.38 | 87.01 | 5.17 | 95.14 | 91.57 | 100.00 | 50.00 | 89.27 | 0.00 | 98.85 |
| WaNet [29] | 84.58 | 96.49 | N/A | 91.10 | 3.36 | 96.56 | 90.68 | 10.23 | 93.13 | 84.58 | 96.49 | 50.00 | 89.82 | 0.96 | 97.76 |
| Avg | 89.02 | 84.44 | N/A | 89.82 | 27.71 | 78.36 | 88.05 | 22.39 | 80.54 | 89.63 | 72.42 | 56.01 | 90.31 | 47.74 | 68.35 |

| ATTACK | i-BAU [45] | | | EP [48] | | | NPD-TU(Ours) | | | NPD-TP(Ours) | | | NPD(Ours) | | |
|---|---|---|---|---|---|---|---|---|---|---|---|---|---|---|---|
| | ACC | ASR | DER | ACC | ASR | DER | ACC | ASR | DER | ACC | ASR | DER | ACC | ASR | DER |
| BadNets-A2O [11] | 87.69 | 3.13 | 94.28 | 87.56 | 7.28 | 92.15 | 90.54 | 5.98 | 94.23 | 90.06 | 3.06 | 95.51 | 89.52 | 0.10 | 96.71 |
| BadNets-A2A [11] | 86.86 | 2.19 | 88.95 | 89.98 | 68.91 | 57.15 | 90.82 | 20.54 | 81.76 | 90.88 | 6.05 | 89.03 | 89.81 | 2.36 | 90.34 |
| Blended [5] | 88.45 | 51.67 | 70.93 | 90.99 | 22.62 | 86.72 | 90.66 | 0.00 | 97.87 | 91.23 | 0.00 | 98.15 | 90.85 | 9.99 | 92.97 |
| Input-Aware [28] | 89.81 | 78.93 | 57.82 | 88.92 | 3.84 | 95.37 | 89.15 | 96.51 | 50.00 | 89.98 | 0.02 | 97.28 | 87.99 | 2.57 | 95.67 |
| LF [46] | 83.06 | 6.66 | 53.48 | 79.36 | 18.88 | 48.04 | 88.78 | 0.96 | 56.44 | 89.05 | 0.89 | 56.47 | 87.64 | 0.91 | 56.46 |
| SSBA [20] | 85.61 | 12.37 | 88.75 | 89.62 | 85.40 | 54.24 | 90.42 | 0.31 | 97.18 | 90.71 | 0.41 | 97.28 | 88.94 | 4.21 | 94.49 |
| Trojan [24] | 86.40 | 2.69 | 96.07 | 89.42 | 5.23 | 96.31 | 90.35 | 0.00 | 99.39 | 90.81 | 8.89 | 95.18 | 89.79 | 0.86 | 98.68 |
| WaNet [29] | 89.61 | 2.40 | 97.04 | 86.10 | 73.61 | 61.44 | 88.85 | 5.44 | 95.52 | 89.03 | 1.35 | 97.57 | 88.25 | 1.88 | 97.30 |
| Avg | 87.19 | 20.00 | 81.30 | 87.74 | 35.72 | 73.72 | 89.95 | 16.22 | 84.11 | 90.22 | 2.58 | 90.93 | 89.10 | 2.86 | 90.79 |

and OSDN [32]. For the denoising structures, we found that inserting four denoising layers in all four blocks of the backdoored model as in OSAD and OSDN leads to the collapse of the entire network. Hence, we adopt a single denoising layer inserted into only one layer of the backdoored model as NPD. This limitation arises due to the scarcity of training samples available for fine-tuning in the defense scenario. As for the Linear layer, it means we use a linear layer with weights and biases that have the same size as the latent features. Consequently, this structure has fewer parameters compared to ours. We conducted experiments on the CIFAR-10 dataset using the PreAct-ResNet18 network, and the other experimental setup aligns with our method. The defense performance results are shown in Table 13.

From the table, we observe that both OSAD and OSDN successfully maintain performance on clean data. However, the ASR experiences a slight drop in most cases, possibly due to the fact that OSAD and OSDN adopt a residual structure where backdoors cannot be completely eliminated. The two Conv-BN layers structure has a high probability of collapsing directly, which may be attributed to the

insufficient training data for such a multi-layer structure. The Linear structure shows lower ACC and higher ASR. This could be because fewer parameters are not sufficient to remove backdoors, and a random initialization of the inserted layer leads to a decrease in final accuracy. This experiment demonstrates the superiority of the designed Conv-BN structure in defending against backdoors.

Table 13: Defense performance under different structures of inserted layers on CIFAR-10 dataset with PreAct-ResNet18.

| Structure → ATTACK ↓ | OSAD [31] | | OSDN [32] | | Linear | | (Conv-BN)×2 | | Conv-BN (**Ours**) | |
|---|---|---|---|---|---|---|---|---|---|---|
| | ACC | ASR | ACC | ASR | ACC | ASR | ACC | ASR | ACC | ASR |
| BadNets-A2O [11] | 91.68 | 48.43 | 90.47 | 45.26 | 87.75 | 21.12 | 10.05 | 99.56 | 88.93 | 1.26 |
| Blended [5] | 87.93 | 12.66 | 91.74 | 26.87 | 86.98 | 44.36 | 90.51 | 9.06 | 91.18 | 0.41 |
| Input-Aware [28] | 93.53 | 92.78 | 92.32 | 60.25 | 86.85 | 58.51 | 10.07 | 99.56 | 89.57 | 0.11 |
| LF [46] | 92.32 | 27.28 | 88.86 | 67.73 | 88.41 | 91.59 | 88.36 | 0.36 | 90.06 | 0.21 |
| SSBA [20] | 89.77 | 87.46 | 90.53 | 84.79 | 88.81 | 79.49 | 10.10 | 99.61 | 90.88 | 2.77 |
| Trojan [24] | 93.45 | 99.99 | 93.38 | 99.99 | 87.85 | 100.00 | 91.80 | 18.80 | 92.37 | 6.51 |
| WaNet [29] | 93.07 | 97.99 | 92.41 | 85.11 | 87.22 | 96.54 | 10.07 | 100.00 | 91.57 | 0.80 |
| Avg | 91.68 | 66.65 | 91.39 | 67.14 | 87.70 | 70.23 | 44.42 | 60.99 | 90.65 | 1.72 |

# E   Additional experimental results

## E.1   Defense experiments on CIFAR-100 dataset

To show the effectiveness of NPD method on classification tasks with more labels, we complement experiment on CIFAR-100 dataset [16], which has 100 classes in total and each class has 500 images for training and 100 images for testing. As shown in Table 14, the average ACC and ASR of NPD are 65.49%, 1.96%, which indicates its competitiveness in comparison to existing defense approaches and showcases its efficacy in more classes scenarios.

Table 14: Comparison with the state-of-the-art defenses on CIFAR-100 dataset with 5% benign data and 10% poison ratio on PreAct-ResNet18 (%).

| ATTACK | Backdoored | | | NC [35] | | | i-BAU [45] | | | NPD (**Ours**) | | |
|---|---|---|---|---|---|---|---|---|---|---|---|---|
| | ACC | ASR | DER | ACC | ASR | DER | ACC | ASR | DER | ACC | ASR | DER |
| BadNets-A2O [11] | 67.23 | 87.43 | N/A | **66.05** | 0.14 | **93.06** | 60.37 | **0.04** | 90.26 | 64.27 | 0.06 | 92.20 |
| Blended [5] | 69.28 | 99.59 | N/A | **67.55** | 98.94 | 49.46 | 63.75 | **0.79** | 96.64 | 66.30 | 4.43 | 96.09 |
| LF [46] | 68.82 | 94.53 | N/A | **67.67** | 21.48 | 85.95 | 63.85 | **0.18** | 94.69 | 65.37 | 0.22 | **95.43** |
| SSBA [20] | 68.97 | 96.42 | N/A | **67.37** | 78.64 | 58.09 | 63.09 | 28.91 | 80.82 | 65.57 | **0.83** | 96.10 |
| Trojan [24] | 68.93 | 100 | N/A | **66.00** | 0.10 | 98.48 | 63.73 | 0.86 | 96.97 | 65.13 | 1.84 | 97.18 |
| Wanet [29] | 69.83 | 98.46 | N/A | **69.76** | 7.14 | 95.62 | 65.31 | 43.96 | 74.99 | 66.27 | 4.36 | 95.27 |
| Avg | 68.84 | 96.07 | N/A | **67.40** | 34.41 | 80.11 | 63.35 | 12.46 | 89.06 | 65.49 | **1.96** | **95.38** |

## E.2   Defense experiments across various model architectures

In this part, we discuss the selection of the layer to insert the neural polarizer (NP) in a new model and the performance of NPD under different model architectures. In the manuscript, we have investigated the impact of NP insertion in different layers on the PreActResNet18 network. Our empirical study demonstrates that inserting NP in deeper layers effectively removes backdoors, while inserting it in shallower layers may lead to a decrease in benign accuracy and instability. To further validate this finding and show the effectiveness and robustness of NPD across different model architectures, we conducted additional experiments on VGG19, Inception-V3, and Densenet161 networks. As presented in Table 15, inserting NP into the last three layers achieves remarkable performance (high ACC and low ASR), while a decrease in accuracy occurs when shallow layers are selected. This experiment shows that NPD is robust across different model architectures.

Table 15: Performance of neural polarizer insertion in different layers across various network architectures (%).

| Network | Layer → Attack ↓ | Backdoored | | Shallow | | Middle | | Third-to-last | | Penultimate | | Last | |
|---|---|---|---|---|---|---|---|---|---|---|---|---|---|
| | | ACC | ASR | ACC | ASR | ACC | ASR | ACC | ASR | ACC | ASR | ACC | ASR |
| VGG19 | BadNets-A2O [11] | 89.36 | 95.93 | 77.58 | 2.83 | 87.41 | 0.40 | 88.68 | 5.17 | 87.28 | 3.81 | 87.73 | 4.06 |
| | Blended [5] | 90.17 | 99.12 | 77.59 | 17.19 | 88.89 | 25.53 | 89.27 | 9.78 | 89.31 | 7.40 | 87.95 | 5.78 |
| | Input-Aware [28] | 77.67 | 94.58 | 70.31 | 1.47 | 70.99 | 20.37 | 74.30 | 7.13 | 72.59 | 7.77 | 77.11 | 1.47 |
| | SSBA [20] | 89.48 | 91.86 | 72.76 | 0.67 | 88.45 | 20.20 | 89.05 | 10.53 | 88.63 | 6.12 | 88.70 | 4.31 |
| Inception-V3 | BadNets-A2O [11] | 90.68 | 94.50 | 11.37 | 92.84 | 90.76 | 10.82 | 91.49 | 8.54 | 90.62 | 2.51 | 91.69 | 4.70 |
| | Blended [5] | 93.73 | 99.82 | 10.00 | 100.00 | 92.70 | 69.77 | 93.71 | 8.91 | 93.05 | 4.51 | 93.71 | 0.01 |
| | Input-Aware [28] | 91.60 | 98.80 | 19.22 | 0.38 | 90.52 | 10.50 | 91.65 | 2.46 | 91.04 | 5.30 | 91.58 | 0.69 |
| | SSBA [20] | 93.33 | 98.30 | 10.00 | 100.00 | 92.21 | 89.81 | 93.35 | 7.52 | 92.88 | 6.13 | 92.23 | 1.13 |
| DenseNet161 | BadNets-A2O [11] | 84.33 | 89.68 | 72.78 | 44.22 | 81.24 | 80.09 | 81.32 | 9.19 | 82.46 | 8.48 | 82.06 | 1.91 |
| | Blended [5] | 86.37 | 98.79 | 75.09 | 3.79 | 82.87 | 85.98 | 82.71 | 13.83 | 83.75 | 9.16 | 83.18 | 4.22 |
| | Input-Aware [28] | 84.46 | 94.41 | 77.20 | 17.11 | 83.65 | 43.57 | 82.81 | 3.56 | 82.02 | 1.76 | 83.44 | 1.56 |
| | SSBA [20] | 84.18 | 84.13 | 70.99 | 9.57 | 81.57 | 30.27 | 82.72 | 7.09 | 82.06 | 8.41 | 81.41 | 8.48 |
| | Avg | 87.95 | 94.99 | 53.74 | 32.51 | 85.94 | 40.61 | 86.76 | 7.81 | 86.31 | 5.95 | 86.73 | 3.19 |

## E.3 Visualization of targeted adversarial perturbation

We empirically establish that targeted adversarial perturbation (T-AP) is a better surrogate to the trigger compared to untargeted adversarial perturbation (U-AP), as demonstrated through quantitative experiments in the manuscript. It could also be explained from the following aspects: From an intuitive standpoint, T-AP can be regarded as a subregion of U-AP, and the unknown trigger is a specific instance of T-AP. Consequently, employing the projected gradient descent (PGD) adversarial attack method, the generated T-AP is more likely to exhibit proximity to the trigger when contrasted with the generated U-AP. Moreover, we visualize the trigger, as well as the generated T-AP and U-AP. As depicted in Fig. 4, the T-AP is more visually similar to the trigger. To reinforce our argument quantitatively, we calculate the $L_2$ distance in the latent space of the backdoored model among the targeted adversarial example (AE), untargeted AE, and the poisoned sample with the trigger. The distance between targeted AE and poisoned sample 36.4 is substantially smaller than that between Untargeted AE and poisoned sample 93.84. This supplementary experiment reaffirms our contention that T-AP serves as a more faithful surrogate to the concealed trigger in comparison to U-AP.

## E.4 Grad-CAM Visualization

We employ Grad-CAM figures to visualize the effects of our NPD on BadNets, Blended, and Trojan attacks, given that these three attack methods employ visible triggers. As illustrated in Figures 5 to 7, Neural Polarizer (NP) effectively redirects the network's attention away from the triggers and toward the subject of figures. This visual evidence substantiates the successful removal of backdoors by NP in backdoored models.

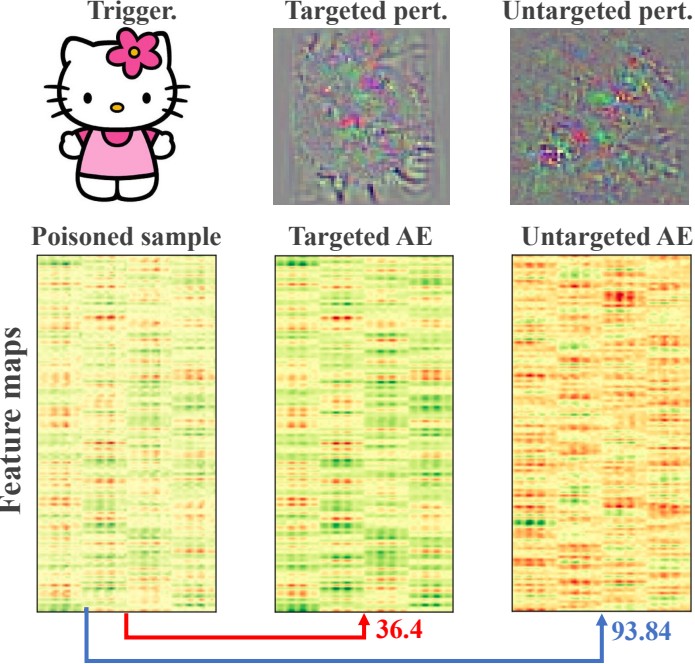

$L_2$ distance of feature maps between AEs and poisoned samples

Figure 4: Comparison of trigger (first column), targeted perturbation (second column), and untargeted perturbation (third column) in the input and latent space. Top left: The trigger used in Blended attack. Top middle: Visualization of the perturbation generated by our targeted adversarial perturbation in NPD. Top right: Visualization of the perturbation generated by the untargeted adversarial perturbation. Bottom left: The feature map derived from the poisoned sample in the feature space. Bottom middle: Feature map of the adversarial example generated through our NPD approach (Targeted AE). Bottom right: Feature map of the adversarial example produced through the untargeted adversarial perturbation (Untargeted AE).

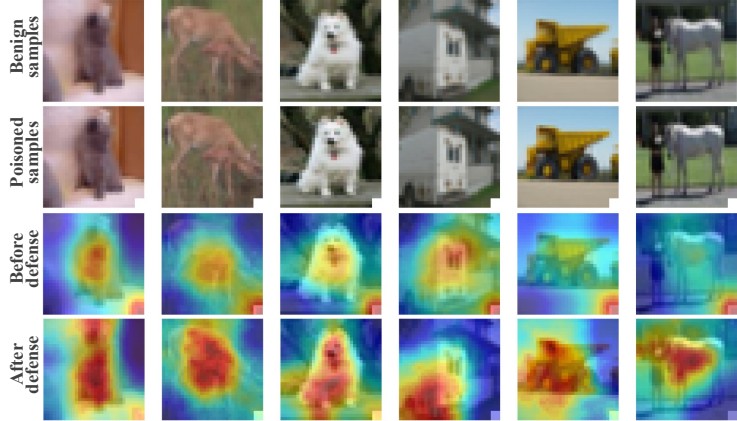

Figure 5: Visualizations of benign and poisoned samples, alongside their corresponding Grad-CAM visualizations before and after the insertion of NP of BadNets attack on CIFAR-10 dataset and PreActResNet18 network.

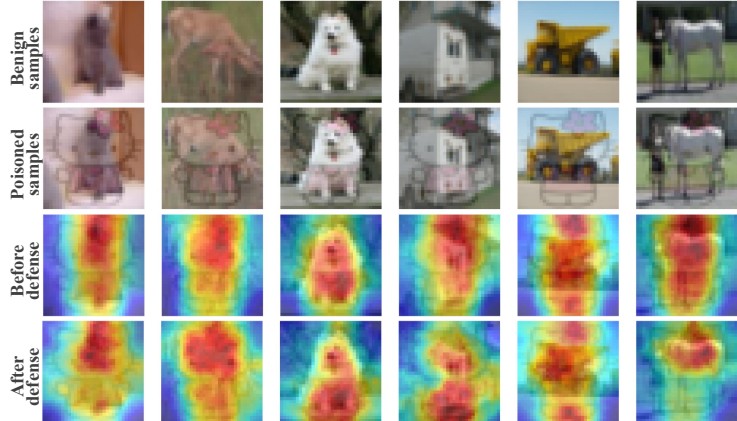

Figure 6: Visualizations of benign and poisoned samples, alongside their corresponding Grad-CAM visualizations before and after the insertion of NP of Blended attack on CIFAR-10 dataset and PreActResNet18 network.

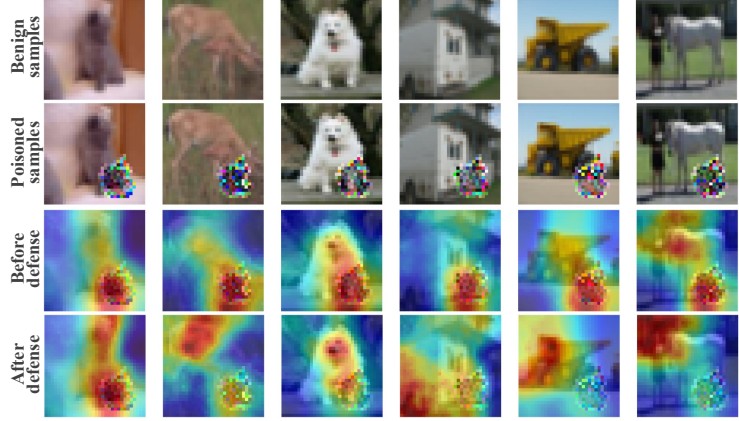

Figure 7: Visualizations of benign and poisoned samples, alongside their corresponding Grad-CAM visualizations before and after the insertion of NP of Trojan attack on CIFAR-10 dataset and PreActResNet18 network.

