# OpenReview forum: "Neural Polarizer: A Lightweight and Effective Backdoor Defense via Purifying Poisoned Features"
_NeurIPS.cc/2023/Conference — NeurIPS 2023 poster_

### Official Review · Reviewer_DJTs · 2023-06-27

**Soundness:** 3 good
**Presentation:** 3 good
**Contribution:** 2 fair
**Rating:** 6
**Confidence:** 5

**Summary:**

This work proposes a defense method against backdoor attacks. The proposed method involves inserting a trainable transformation layer inside a backdoor model while keeping other model parameters fixed that is supposed to purify the poisonous features while allowing benign features to pass without modification.

**Strengths:**

1)	The proposed transformation block constitutes a single 1x1 convolution and batch normalization layer which makes it easier to adopt in any model architecture.
2)	Faster training time since only the added layer is trained while keeping other model parameters fixed.
3)	The proposed method provides good defense performance (low ASR) against various attacks.


**Weaknesses:**

1)	To filter out poisonous features from a poisonous sample, the method requires the creation of poisonous samples. While the authors claim to approximate the trigger, they utilize adversarial samples instead of poisoned ones, which are fundamentally different and not equivalent approximations.
2)	Previous research [41] has utilized adversarial perturbation to address removing backdoor attacks. It is crucial to emphasize the key differentiation between [41] and the proposed NPD-TU method, aside from the difference in training the entire model versus solely training the added transformation layer.
3)	The proposed method exhibits a decrease in benign accuracy, as indicated by the results presented in Table 2. This decline in accuracy is a common occurrence observed in adversarial training techniques.
4)	The authors insert the transformation block before the third convolution layer of the fourth layer for PreAct-ResNet18. How is this design decision made, i.e., for a new model which might be potentially backdoored? How does the defender decide where to insert this polarizer block?
5)	How does the proposed method perform under an adaptive attack, where the attacker knows the defense strategy? One approach to testing this could be training the backdoor model using adversarial samples. It would be interesting to see this result.
6)	How does the proposed method affect the performance of a benign model?


**Questions:**

1)	How compatible is neural polarizer with other model architectures like inception and Densenet?
2)	How is the defense performance against all2all attacks for attacks other than BadNets?


**Limitations:**


I have raised some concerns in the weaknesses section. Those are some possible limitations of the work. The authors should revise their limitation section.

---

> ### Author Rebuttal · Authors · 2023-08-09
>
> Thank you for your dedicated time reading our paper and providing us with your meticulous review. Your insightful questions and concerns are greatly appreciated. Please inform us if these responses effectively address all your inquiries or if there are additional questions you'd like to raise.
>
> **Q1. Utilize adversarial samples instead of poisoned ones, which are fundamentally different and not equivalent approximations.**
>
> **R1:** Thanks for this insightful concern. It is essentially the same as Q2 proposed by Reviewer qQYz. Due to the space limit, we would like to refer you to **R2 to Reviewer qQYz**.
>
> **Q2. Key differentiation between i-BAU [41] and NPD-TU.**
>
> **R2:** There are still three significant differences between NPD-TU and i-BAU:
> * **Formulation:** NPD-TU has a min-min formulation, while i-BAU has a min-max formulation. The reason is that NPD-TU computes targeted adversarial perturbation (T-AP), while i-BAU computes untargeted AP (U-AP). The benefit of T-AP is explained in the **first response**.
> * **Algorithm:** i-BAU adopts an implicit hypergradient algorithm, involving the inverse of Hessian matrix, which is very costly and difficult to scale to large image size. In contrast, NPD alternatively solves the inner and outer minimization using efficient PGD and SGD algorithm, respectively. The running time of NPD and i-BAU is presented in **Table 1**, which shows NPD is more efficient than i-BAU, especially on Tiny ImageNet.
> * **Performance**: As shown in Tables 1 and 2 in manuscript, i-BAU performs poorly on defending against several attacks (especially on Tiny ImageNet dataset). In contrast, our NPD-TU and NPD show better defense performance across all attacks.
>
> **Table 1: Running time (sec.) in comparison with state-of-the-art defenses with 2500 images on PreAct-ResNet18.**
> |Defense|FP|NAD|NC|ANP|i-BAU|NPD|
> :-:|:-:|:-:|:-:|:-:|:-:|:-:
> CIFAR-10|1169.01|74.39|896.45|58.75|57.23|55.16
> Tiny ImageNet|3357|351|42512|1692|887|332
>
> **Q3. Where to insert this polarizer block for a new backdoored model?**
>
> **R3:** Thanks. Due to space limit, we refer you to **R2 of Reviewer BztP** for more details, which contains an in-depth analysis and performance of NP insertion in various layers across VGG19, Inception-V3, and Densenet161 networks, respectively. Briefly, NPD is robust across different network structures. And, it remains effective and reliable when the NP is inserted into deep layers of the network.
>
> **Q4. Defend against adaptive attacks.**
>
> **R4:** Thanks for this constructive suggestion. We train the backdoored models with adversarial training (AT) to serve as the adaptive attack. The defense performance against the AT backdoored models is shown in **Table 2**.
> * **NPD still performs well on AT models**, while i-BAU performs poor. The possible reason is that i-BAU is essentially adversarial training, while NPD adopts dynamic targeted adversarial perturbation, which is different from AT.
> * **Compared to the defense against backdoored models with standard training** (see Table 1 in main manuscript), there is a slight ASR increase from 1.62% to 5.22%. However, AT models also sacrifices the ACC.
>
> **Table 2: Defense performance of NPD against adaptive attacks on CIFAR-10.**
>
> ||Backdoored||i-BAU||NPD||
> -|:-:|:-:|:-:|:-:|:-:|:-:
> ||ACC|ASR|ACC|ASR|ACC|ASR
> Blended|86.00|99.63|83.98|30.43|83.64|3.92
> Input-Aware|84.98|94.99|83.17|71.12|83.13|4.47
> LF|84.15|94.30|84.15|94.30|83.39|4.89
> SSBA|84.34|93.32|82.74|24.54|83.38|5.22
>
> **Q5. Influence on benign models.**
>
> **R5:** Our NPD defense on benign models trained on different datasets is presented in Table 3 ('PreAct' indicates Pre-ActResNet18 and 'VGG' indicates VGG19-BN). It shows that there is a slight influence on benign models.
>
> **Table 3: Defense performance on benign models.**
> ||CIFAR-10||GTSRB||Tiny ImageNet||
> :-:|:-:|:-:|:-:|:-:|:-:|:-:
> ||PreAct|VGG|PreAct|VGG|PreAct|VGG
> Benign Model|93.70|92.07|98.15|98.11|57.35|47.18
> After Defense|92.58|91.00|98.43|98.06|52.15|47.01
>
> **Q6. How compatible is neural polarizer with other model architectures like Inception and DenseNet?**
>
> **R6:** The evaluations of NPD and other SOTA defenses against different attacks across Inception-V3 and DenseNet161 are shown in **Table 4**. Our NPD shows superior performance across the two networks, demonstrating the robustness of NPD under different model architectures.
>
> **Table 4: Defense results in comparison with NC and i-BAU on Inception-V3 and DenseNet161.**
>
> |||Backdoored||NC||i-BAU||NPD||
> -|:-:|:-:|:-:|:-:|:-:|:-:|:-:|:-:|:-:
> |||ACC|ASR|ACC|ASR|ACC|ASR|ACC|ASR
> Inception-V3|BadNets|90.68|94.50|89.87|1.21|82.41|0.52|91.69|4.70
> ||Blended|93.73|99.82|93.73|99.82|83.63|8.84|93.71|0.01
> ||Input-Aware|91.60|98.80|91.60|98.80|91.03|22.93|91.58|0.69
> ||SSBA|93.33|98.30|91.56|50.73|84.74|1.92|92.23|1.13
> DenseNet161|BadNets|84.33|89.68|82.69|2.82|79.49|39.63|82.06|1.91
> ||Blended|86.37|98.79|86.38|98.79|77.86|48.44|83.18|4.22
> ||Input-Aware|84.46|94.41|84.45|94.41|81.96|17.14|83.44|1.56
> ||SSBA|84.18|84.13|83.26|14.50|80.51|11.16|81.41|8.48
>
> **Q7. Defense performance against all2all attacks.**
>
> **R7:** Thanks for this constructive suggestion. The dynamic and sample-specific prediction strategy of target label enables NPD to defend against all2all attacks. We conduct a supplementary experiment on CIFAR-10 dataset. The low ASR in **Table 5** tells that NPD can successfully defend against all2all backdoor attacks, showing the adaptation and robustness of our method.
>
> **Table 5: Defense performance against all2all attacks on CIFAR-10 dataset with 5% benign data and 10% poisoning ratio on PreAct-ResNet18 (%).**
> ||Backdoored||NPD||
> :-:|:-:|:-:|:-:|:-:
> ||ACC|ASR|ACC|ASR
> Blended|91.59|83.50|91.07|6.24
> LF|86.60|78.55|90.10|2.10
> Input-Aware|91.91|84.80|90.53|4.88
> SSBA|91.30|85.04|91.14|1.29

---

> ### Author Response · Authors · 2023-08-16
> **Seeking Your Valuable Feedback**
>
> Dear Reviewer **DJTs**,
>
> We would like to extend our appreciation for your time and valuable comments. We are eagerly looking forward to receiving your valuable feedback and comments on the points we addressed in the rebuttal. Ensuring that the rebuttal aligns with your suggestions is of utmost importance.
>
> Thank you for your dedication to the review process.
>
> Sincerely,
>
> Authors

---

> > ### Comment · Reviewer_DJTs · 2023-08-17
> >
> > Thanks for such a detailed response. The authors have addressed most of my concerns with detailed experimental results. Thus I
> > have increased my score from borderline accept to weak accept.

---

> > > ### Author Response · Authors · 2023-08-17
> > > **Thanks for your feedback**
> > >
> > > Dear Reviewer DJTs,
> > >
> > > We sincerely appreciate your thoughtful response and the time you've dedicated to reviewing our paper. We will incorporate your suggestions and insights into the revised manuscript. Thank you once again for your thorough review and your positive evaluation. Your support and input are greatly appreciated.
> > >
> > > Sincerely,
> > >
> > > Authors

---

### Official Review · Reviewer_qQYz · 2023-07-04

**Soundness:** 3 good
**Presentation:** 3 good
**Contribution:** 2 fair
**Rating:** 7
**Confidence:** 4

**Summary:**

This paper proposes propose a backdoor defense method by inserting a learnable neural polarizer into the backdoored model as
an intermediate layer, in order to purify the poisoned sample via filtering trigger information while maintaining benign information. To more effectively remove backdoor, this paper leverages the reverse trigger and target label to remove backdoor.

**Strengths:**

1. This paper is well written and technically sound.
2. The experiments are sufficient, and its performance achieves SOTA

**Weaknesses:**

1. Neural Polarizer needs to reverse the backdoor trigger and the target label, which is similar to Neural Cleanse. Furthermore, former papers demonstrate that trigger reverse mainly works well on static trigger such as BadNets. The results on sample-specific trigger are bad and the results of reverse trigger and target label are mostly wrong. Thus, the scope where Neural Polarizer can be used is limited.
2. The novelty is limited. Because trigger reversed has been proposed before, the contribution of this paper is to finetune the model with the reversed trigger on Neural Polarizer layer.

**Questions:**

Can you explain why the method works with the reversed trigger which may not be correctly reversed?

**Limitations:**

Yes

---

> ### Author Rebuttal · Authors · 2023-08-09
>
> Thank you for your valuable time in reading our work and positive review on our techniques and experimental results. We appreciate your insightful questions and comments.
>
> **Q1. Neural Polarizer (NP) needs to reverse the backdoor trigger and the target label, which is similar to Neural Cleanse. The scope where NP can be used is limited. The novelty is limited.**
>
> **R1:** Although both our NPD and NC employ adversarial techniques and targeted label to fine-tune the backdoored model, our NPD shows novelty and differs from NC in the following three aspects:
>
> **1. Dynamic and sample-specific prediction strategy of target label.** We predict the target label for each training sample by $y_i^{'} = \arg\max_{k_i \neq y_i} \hat{f}_{k_i}(\mathbf{x}_i, \boldsymbol{\theta})$, while NC estimates a common target label for all samples. It has three strengths:
> * **The predicted target label enables us to generate targeted adversarial perturbation (T-AP), which shows better performance compared to untargeted adversarial perturbation(U-AP)**. The reason is that T-AP is more effective as surrogate perturbations to the triggers. We have provided ablation study in Table 4 in the manuscript. We also provide a comparison of generated T-AP, U-AP, and corresponding poisoned sample in the latent space of a backdoor model in **supplementary PDF**, which shows T-AP is more visually similar to poisoned sample than U-AP.
> * **The dynamic and sample-specific target label prediction is applicable to both all2one and all2all attack settings** (see **Table 1** for defense on all2all attacks). Since the defender doesn’t know whether the attack is all2one, all2all, or has multi-trigger multi-target, our strategy can flexibly cope with these situations and plays a better false tolerance role.
>
> **2. Targeted and sample-specific adversarial perturbation.** Compared to targeted universal adversarial perturbation and the local mask strategy used in NC, our NPD adopts the targeted and sample-specific adversarial perturbation (AP) and doesn't restrict the format of AP. Therefore, NPD can cope with various types of backdoor triggers, e.g., sample-specific, or global. This is also reflected in our experimental performance.
>
> **3. Fine-tuning Paradigm.** In comparison to NC which needs to fine-tune the whole network, our NPD only fine-tunes the inserted transformation layer. It is **data-efficient** which performs well with very few clean data, and **computationally efficient**, which needs less running time for defense. Please refer to Tables 6 and 7 in the manuscript for experimental details.
>
> A systematic and detailed analysis of NPD's mechanism and novelty are presented in the first **Common Response**.
>
> **Table 1: Defense performance under various all2all attacks on CIFAR-10(%).**
> ||Backdoored||NPD||
> :-:|:-:|:-:|:-:|:-:
> ||ACC|ASR|ACC|ASR
> Blended|91.59|83.50|91.07|6.24
> LF|86.60|78.55|90.10|2.10
> Input-Aware|91.91|84.80|90.53|4.88
> SSBA|91.30|85.04|91.14|1.29
>
> **Q2. Why does the method work with the reversed trigger which may not be correctly reversed?**
>
> **R2:** Thanks for this insightful concern. Since it is difficult to access or exactly recover the unknown trigger, one feasible solution is to find some surrogates or approximations, such as AP in our NPD, i-BAU and NC. Moreover, we argue that the targeted AP (T-AP) adopted in NPD is a good surrogate to the trigger and better than the untargeted AP (U-AP), in three ways:
> * **Intuitively, T-AP is more likely to be close to the trigger than U-AP.** Because, the set of T-AP is a sub-region of that of U-AP, and the trigger is a particular T-AP.
> * **We visualize and quantitatively measure the distance of adversarial example with T-AP (i.e., targeted AE) and poisoned sample in Fig.1 in supplementary PDF.** It shows that T-AP is more visually similar to the trigger than U-AP. The distance between targeted AE and poisoned sample in the latent space of backdoored model is much smaller than that between untargeted AE and poisoned sample. It verifies that T-AP serves as a better surrogate to poisoned sample.
> * **T-AP makes significant contribution to backdoor defense.** To investegate the effects of T-AP and U-AP on backdoor defense, we compare NPD with two variants NPD-UP and NPD-UU (see Table 4 in manuscript). It shows an obvious gap between NPD and NPD-UP, i.e., the average ASR 1.99% vs. 8.32%. It demonstrates that T-AP makes significant contribution to the success of NPD.
>
> Hope above points could address your concern. Thanks again for your constructive comment.

---

> > ### Comment · Reviewer_qQYz · 2023-08-19
> >
> > Thanks for your detailed rebuttal and most of concerns are solved. Thus, I raise my scores.

---

> > > ### Author Response · Authors · 2023-08-19
> > > **Thank you for your feedback**
> > >
> > > Dear Reviewer qQYz,
> > >
> > > Thank you very much for your time and for your thoughtful feedback. We sincerely appreciate your diligence in evaluating our work. We are pleased to learn that our detailed rebuttal has addressed most of your concerns and has positively influenced your assessment of the manuscript. All suggested experiments and analysis will be added into the revised manuscript.
> > >
> > > Thank you again for your time, consideration, and invaluable feedback.
> > >
> > > Sincerely,
> > > Authors

---

> ### Author Response · Authors · 2023-08-16
> **Seeking Your Valuable Feedback**
>
> Dear Reviewer **qQYz**,
>
> We wish to express our gratitude for your dedicated time and insightful comments. We are anxiously awaiting your valuable feedback and insights regarding the points we addressed in the rebuttal. Ensuring your satisfaction with our rebuttal is of utmost importance to us.
>
> We sincerely appreciate your commitment to the review process and value the time.
>
> Sincerely,
>
> Authors

---

### Official Review · Reviewer_BztP · 2023-07-07

**Soundness:** 3 good
**Presentation:** 3 good
**Contribution:** 3 good
**Rating:** 6
**Confidence:** 4

**Summary:**

The paper proposes a lightweight and effective backdoor defense by inserting a trainable neural layer block. Without modifying original backdoor model, the proposed method can remove backdoor behaviors by filtering poisoned features via the trainable neural block. The authors conduct sufficient experiments to demonstrate the effectiveness of proposed method.

**Strengths:**

1. The proposed method is simple but effective. The inserted neural block only includes one convolution layer followed by a BN layer. Training this layer block is not time-consuming.
2. The experiments are very sufficient. The comparison experiments are conducted with six backdoor defenses against seven backdoor attacks on three datasets. Analysis experiments are also conducted including the effectiveness of losses, poisoning ratios and clean ratios.

**Weaknesses:**

1. The paper does not introduce how to choose different layers to insert the neural polarizer. Since the proposed method largely depends on network architectures. Please also provide more results using more network architectures e.g. vgg.

**Questions:**

1. Does this proposed method work for a network without batch normalization layer?
2. Please explain more about how to choose layers to insert neural polarizer.
3. Could the authors visualize features before and after neural polarizer to qualitatively analyze the effectiveness (e.g. using GradCam)?

**Limitations:**

The authors have addressed the limitations

---

> ### Author Rebuttal · Authors · 2023-08-09
>
> We sincerely thank the reviewer for carefully reading our paper. We are encouraged by the positive comments of simple but effective method and sufficient experiments.
>
> **Q1. More results using more network architectures, e.g., vgg. Does this proposed method work for a network without batch normalization layer?**
>
> **R1:** We appreciate your valuable question. We perform defense results of NPD in comparison with NC and i-BAU in **Table 1**. It shows that NPD works for VGG19, which doesn't have batch normalization layer. Please refer to **Table 2 in R2** for more experimental results, which contains performance of NPD under more architecture, i.e., Inception-V3, and Densenet161 networks. In brief, our NPD approach demonstrates robustness and effectiveness across a wide range of network architectures.
>
> **Table 1: Performance of NPD in comparison with NC and i-BAU on VGG19 network.**
>
> ||Backdoored||NC||i-BAU||NPD||
> :-:|:-:|:-:|:-:|:-:|:-:|:-:|:-:|:-:
> |ATTACK|ACC|ASR|ACC|ASR|ACC|ASR|ACC|ASR
> BadNets|89.36|95.93|87.90|49.66|14.61|0.43|87.73|4.06
> Blended|90.17|99.12|89.09|96.72|88.61|59.86|87.95|5.78
> Input-Aware|77.67|94.58|74.45|4.02|72.15|16.22|77.11|1.47
> SSBA|89.48|91.86|89.48|91.86|88.08|3.74|88.70|4.31
>
> **Q2. How to choose layers to insert neural polarizer.**
>
> **R2:** Thanks and we acknowledge that the selection of layer is important for NP. Actually we have investigated the performance of NP insertion into different layers on PreAct-ResNet18 in manuscript. To further investigate the impact of layer selection, we conduct experiments on three more backbones in **Table 2**, i.e., VGG19, Inception-V3, and Densenet161 networks, besides PreAct-ResNet18 and VGG19-BN in our paper. From Table 2 below and Figure 4 in our paper, we find that: inserting NP into the last three layers achieves remarkable performance (high ACC and low ASR), while a decrease in accuracy occurs when shallow layers are selected.
>
> Therefore, the last feature layer is recommended, which yields better defense performance and maintains robustness against backdoor attacks.
>
> **Table 2: Performance of neural polarizer insertion in different layers across various network architectures(%).**
> |||Shallow||Middle||Third-to-last||Penultimate||Last||
> :-:|:-:|:-:|:-:|:-:|:-:|:-:|:-:|:-:|:-:|:-:|:-:
> |||ACC|ASR|ACC|ASR|ACC|ASR|ACC|ASR|ACC|ASR
> VGG19|BadNets|77.58|2.83|87.41|0.40|88.68|5.17|87.28|3.81|87.73|4.06
> ||Blended|77.59|17.19|88.89|25.53|89.27|9.78|89.31|7.40|87.95|5.78
> ||Input-Aware|70.31|1.47|70.99|20.37|74.30|7.13|72.59|7.77|77.11|1.47
> ||SSBA|72.76|0.67|88.45|20.20|89.05|10.53|88.63|6.12|88.70|4.31
> Inception-V3|BadNets|11.37|92.84|90.76|10.82|91.49|8.54|90.62|2.51|91.69|4.70
> ||Blended|10.00|100.00|92.70|69.77|93.71|8.91|93.05|4.51|93.71|0.01
> ||Input-Aware|19.22|0.38|90.52|10.50|91.65|2.46|91.04|5.30|91.58|0.69
> ||SSBA|10.00|100.00|92.21|89.81|93.35|7.52|92.88|6.13|92.23|1.13
> DenseNet161|BadNets|72.78|44.22|81.24|80.09|81.32|9.19|82.46|8.48|82.06|1.91
> ||Blended|75.09|3.79|82.87|85.98|82.71|13.83|83.75|9.16|83.18|4.22
> ||Input-Aware|77.20|17.11|83.65|43.57|82.81|3.56|82.02|1.76|83.44|1.56
> ||SSBA|70.99|9.57|81.57|30.27|82.72|7.09|82.06|8.41|81.41|8.48
>
> **Q3. Visualize features before and after neural polarizer by Grad-CAM.**
>
> **R3:** To better understand the effect of neural polarizer (NP) in purifying poisoned features, we visualize the benign samples, poisoned samples, and their Grad-CAM visualization before and after NP in **Fig. 2-4 in supplementary PDF**. We visualize BadNets, Blended, and Trojan attacks since they use visible triggers. As the figures show, NP corrects the network's attention from triggers to the subject of figures, showing the NP successfully removes backdoors in backdoored models.

---

> > ### Comment · Reviewer_BztP · 2023-08-15
> >
> > I have read the rebuttal, and most of my concerns are solved. Thanks!

---

> > > ### Author Response · Authors · 2023-08-16
> > > **Thanks for your feedback**
> > >
> > > Dear Reviewer BztP,
> > >
> > > Thanks for your feedback. We are strongly encouraged by your recognition of our efforts. All suggested experiments and analysis will be added into the revised manuscript. Thanks again for your valuable time and constructive comments.
> > >
> > > Sincerely,
> > > Authors

---

### Official Review · Reviewer_cjy4 · 2023-07-20

**Soundness:** 2 fair
**Presentation:** 3 good
**Contribution:** 2 fair
**Rating:** 6
**Confidence:** 3

**Summary:**

This paper proposes a novel defense method to filter trigger information from poisoned samples. It inserts a learnable intermediate layer (called neural polarizer) into the backdoored model, and proposes bi-level optimization solution to approximate perturbation and target label. Experimental results demonstrate the effectiveness of Neural Polarizer over other defense baselines.

**Strengths:**


1.The paper is well-written and easy to follow.

2.The proposed Nueral Polarizer is interesting.

3.The experiments are comprehensive.



**Weaknesses:**

1.The paper proposes NPD (Neural Polarizer based backdoor Defense) as a normal setting. It also proposes NPD-TP (assume the target label is known) and NPD-TU (assume having full access of benign samples). The NPD-TP and NPD-TU relax the limitation, and is kind of unfair compared to other defense baselines. However, based on Table 1 and Table 2, most of best performance comes from NPD-TP/NPD-TU, while NPD does not perform very well under some situations.

More specific, In Table 1, defense on CIFAR-10: NPD does not always perform the best. For example, ASR under WaNet is 0.80 (larger than WaNet with FP and NAD), ASR under Trojan is 6.51 (larger than Trojan with i-BAU and EP). Similar phenomenon happens in Table 2: defense on Tiny ImageNet.

**Questions:**


1.Regarding approximating perturbation $\Delta$ and target label T (eq.5): in inner minimization, the algorithm first estimates the target label, then applies PGD to generate perturbations. I assume this step is very important, because it decides the outer minimization. That are the estimated perturbations look like? Are they similar with the real triggers, or they are just the adversarial perturbations? Are there any experiments/visualizations to elaborate that?

2.How to choose which layer to insert the neural polarizer? Does your method robust to the choice of layer?

3.What if there are more labels. For example, CIFAR-100 or even more labels. Does the proposed method still work?

**Limitations:**

The authors adequately addressed the limitations.

---

> ### Author Rebuttal · Authors · 2023-08-09
>
> Firstly, we would like to show our sincere appreciation to the reviewer for the valuable time and constructive comments on our submission.
>
> **Q1. The variants NPD-TP (assume the target label is known) and NPD-TU (assume having full access to benign samples) relax the limitation, and is kind of unfair compared to other defense baselines. Most of best performance comes from NPD-TP/NPD-TU, while NPD does not perform very well under some situations.**
>
> **R1:** Thanks. We would like to explain from the following three points.
> * **Assumption of NPD-TP and NPD-TU.** We would like to clarify that NPD-TP and NPD-TU share the same assumption that the target label is known. We think that the sentence at Line 186 "*the targeted universal adversarial perturbation (TUAP) for all benign samples, dubbed NPD-TU*" may mislead the reviewer to think that NPD-TU assumes to have full access to benign samples. Actually, that means the access of all benign samples used for fine-tuning, rather than all benign samples in the original training dataset. That is same with NPD, NPD-TP and all other compared defense methods.
> * **The role of NPD-TP and NPD-TU.** The intrinsic difference between NPD and NPD-TP/NPD-TU is the target label: the former obtains the target label via a dynamic strategy of target label prediction, while the latter assumes to have the true target label. Thus, NPD-TP/NPD-TU serves as the reference to study the effect of the dynamic prediction strategy in NPD. We would like to refer you to the analysis of its effect and mechanism in the **first common response** posted at the top position.
> * **Performance of NPD.** If removing NPD-TP and NPD-TU from Tables 1 and 2 in the main manuscript, we can see that NPD still has a significant performance advantage over all other compared defense methods. Specifically, in Table 1, the most competitive method is i-BAU, its average ACC/ASR/DER are 88.95/6.11/92.42%, while those of NPD are 90.75/1.62/95.56%; in Table 2, the most competitive method is ANP, its average ACC/ASR/DER are 42.64/15.87/80.03%, while those of NPD are 50.55/2.17/90.84%. Although NPD doesn't perform the best at all cases, we believe the huge advantage of the overall performance is enough to demonstrate the effectiveness of NPD.
>
> Hope above points could address your concern, and we will clearly clarify the assumption and role of NPD-TP and NPD-TU in the revised manuscript. Thanks again for your constructive comment.
>
> **Q2. What do the estimated perturbations look like? Are they just adversarial perturbations?**
>
> **R2:** Thank you for your insightful question. We would like to refer you to the **common response** for a comprehensive analysis of the mechanism of NPD, where **visualization of the estimated perturbations** is provided in **Fig. 1** in the **supplementary PDF**.
>
> More importantly, we would like to briefly summarize some important information for your reference. In the visualization, both targeted adversarial perturbation (T-AP) and untargeted adversarial perturbation (U-AP), as well as the corresponding adversarial examples (T-AE, U-AE) are provided. To explain the effect of T-AP, we also present the $L_2$ distance between T-AE, U-AE, and poison samples in the latent space. From the visualization, we find that
>
> 1. **Visualization:** T-AP exhibits more visual similarity to the trigger both in the input space and latent space compared to U-AP, showing targeted perturbations are more effective as surrogate perturbations to the triggers.
>
> 2. **Quantitative analysis:** In latent space, the $L_2$ distance between T-AE and the poisoned sample is 36.4, which is significantly smaller than the distance between U-AE and the poisoned sample, i.e., 93.84. This quantitative analysis further supports the efficacy of our targeted adversarial perturbations as a better surrogate for unknown triggers.
>
> Overall, both visualization and quantitative analysis provide substantial evidence to validate the effectiveness of NPD in producing targeted adversarial perturbations, which is different from ordinary adversarial perturbations. Hope the above points could address your questions.
>
> **Q3. How to choose the layer to insert the neural polarizer? Does your method robust to the choice of layer?**
>
> **R3:** We appreciate your question. Due to space limit, we refer you to **R2 of Reviewer BztP** for more details, which contains an in-depth analysis and performance of NP insertion in various layers across VGG19, Inception-V3, and Densenet161 networks, respectively. Briefly, NPD is robust across different network structures. And, it remains effective and reliable when the NP is inserted into deep layers of the network.
>
> **Q4. What if there are more labels, e.g., CIFAR-100?**
>
> **R4:** Thanks for this suggestion. Actually, we have evaluated NPD on Tiny ImageNet, which has 200 classes. The results on CIFAR-100 in comparison with two SOTA defenses are presented in **Table 1**. The average (ACC, ASR) of NPD is (65.49%, 1.96%), while those of NC and i-BAU are (67.4%, 34.41%) and (63.35%, 12.46%), respectively. Our NPD shows significant advantages.
>
> **Table 1: Comparison with the state-of-the-art defenses on CIFAR-100 dataset with 5% benign data and 10% poisoning ratio on PreAct-ResNet18 (%).**
> ||Backdoored||NC||i-BAU||NPD||
> :-:|:-:|:-:|:-:|:-:|:-:|:-:|:-:|:-:
> ||ACC|ASR|ACC|ASR|ACC|ASR|ACC|ASR
> BadNets|67.23|87.43|66.05|0.14|60.37|0.04|64.27|0.06
> Blended|69.28|99.59|67.55|98.94|63.75|0.79|66.30|4.43
> LF|68.82|94.53|67.67|21.48|63.85|0.18|65.37|0.22
> SSBA|68.97|96.42|67.37|78.64|63.09|28.91|65.57|0.83
> Trojan|68.93|100.00|66.00|0.10|63.73|0.86|65.13|1.84
> Wanet|69.83|98.46|69.76|7.14|65.31|43.96|66.27|4.36
> Avg|68.84|96.07|67.40|34.41|63.35|12.46|65.49|1.96

---

> > ### Comment · Reviewer_cjy4 · 2023-08-14
> > **Thanks for your response.**
> >
> > Thanks for the author's response. You have released most of my concern. I have increased the score.

---

> > > ### Author Response · Authors · 2023-08-15
> > > **Thank you for your response.**
> > >
> > > Thank you very much for your response and kind words. We are truly appreciative to hear that most of your concerns have been addressed. We will add above analysis into the revised manuscript. Thanks again for your valuable time and comments.

---

### Author Rebuttal · Authors · 2023-08-09

# Common Response
We sincerely thank all reviewers for their time and constructive comments.

**Q1. Analysis of the mechanism and novelty of our neural polarizer defense (NPD) method.**

**R1：** We aim to present **a systematic analysis** about NPD's mechanism and novelty covering **three critical components**:

**1. Dynamic and sample-specific target label prediction strategy**
* **Definition and contrast with existing methods.** We estimate the target label for each training sample in inner-minimization step (See Line 177-178 in manuscript). **In contrast**, Neural Cleanse (NC) estimates a common target label for all samples.
* ***Target label prediction accuracy.*** In **Table 1**, we present accuracies of dynamic target label prediction on backdoor attacks at the first mini-batch. Most prediction accuracies are much higher than random guess (10\%). Although the predicted target label isn't 100% correct, it remains effective for backdoor removal, as analyzed later.
* **Effect & analysis.** There are two main advantages to the dynamic strategy:
  * **The predicted target label enables us to generate targeted adversarial perturbation (T-AP).** As analyzed later, T-AP is a better surrogate for the unknown trigger than the untargeted adversarial perturbation (U-AP), enhancing backdoor removal efficacy. Despite the partial accuracy of predicted target labels, they enable robust backdoor defense. Incorrectly predicted labels guide us to generate U-AP (aimed at other classes). Thus, the generated adversarial perturbations of all fine-tuning samples in NPD are a **mixture of T-AP and U-AP**.
  * **The sample-specific target label prediction is applicable to all2one and all2all attack settings.** In practice, defenders lack certainty about attack types (all2one or all2all), risking suboptimal defenses due to erroneous guesses or detection. Our sample-specific target label prediction avoids this risk. Thus, our method performs well against both all2one and all2all attacks (see **Table 2**).

**Table 1: Accuracy of dynamic target label prediction on CIFAR-10(%).**
|ATTACK|BadNets|Blended|Input-Aware|LF|SSBA|Trojan|Wanet|Avg|
:-:|:-:|:-:|:-:|:-:|:-:|:-:|:-:|:-:
ACC|86.33|33.05|72.66|43.75|37.11|28.91|84.77|55.22

**Table 2: Defense performance under different all2all attacks on CIFAR-10(%).**
||Backdoored||NPD||
:-:|:-:|:-:|:-:|:-:
||ACC|ASR|ACC|ASR
BadNets|91.89|74.42|91.41|0.89
Blended|91.59|83.50|91.07|6.24
Input-Aware|91.91|84.80|90.53|4.88
SSBA|91.30|85.04|91.14|1.29

**2. Targeted and sample-specific adversarial perturbation**
* **Definition and contrast with existing methods.** We generate T-AP for each training sample in the inner-minimization step (Line 178-179 and Eq. (5) in manuscript). **In contrast**, i-BAU generates U-AP. NPD's min-min formulation contrasts i-BAU's min-max one. Besides, NC adopted targeted universal adversarial perturbation in a local patch to approximate the unknown trigger.
* **Effect & analysis.**
  * **T-AP is a better surrogate to the unknown trigger than U-AP:**
    * **Intuitively**, T-AP set is a sub-region of U-AP set, and the unknown trigger is a particular T-AP. Thus, using the projected gradient descent (PGD) adversarial attack method, the generated T-AP is more likely to be close to the trigger than the generated U-AP.
    * **Visualization.** we visualize the trigger, as well as the generated T-AP and U-AP. As shown in **Fig. 1 in supplementary pdf**, the T-AP is more visually similar to the trigger.
    * **Quantitative distance.** We calculate $L_2$ distance in the latent space of backdoored model among the targeted adversarial example (AE), untargeted AE, and the poisoned sample with trigger. The distance between Targeted AE and poisoned sample 36.4 is much smaller than that between Untargeted AE and poisoned sample 93.84.

  * **T-AP & U-AP Mixture for Effective Defense.** As shown above, NPD generates a mixture of T-AP and U-AP. In contrast, in the variants NPD-TP (see Line 182-185 in manuscript) and NPD-UP (see Line 259), the generated perturbations are pure T-AP and U-AP, respectively. In Tables 1 and 2 of the main manuscript, there is slight gap between NPD and NPD-TP. In Table 4 of the main manuscript, there is an obvious gap between NPD and NPD-UP, i.e., the average ASR 1.99% vs. 8.32%. These two comparisons demonstrate that a mixture of T-AP and U-AP is enough to achieve good defense performance close to the pure T-AP, and the T-AP part makes significant contribution.

**3. Neural polarizer (NP)**
* **Definition and contrast with existing methods.** NP is a lightweight linear transformation layer (see Section 3.2 and Fig. 3 in manuscript). It's inserted into the backdoored model. Only its parameters are fine-tuned, while the parameters of the original backdoored model are fixed. **In contrast**, most existing fine-tuning based defense methods (e.g., i-BAU, NAD) don't modify the structure and fine-tune the whole model.
* **Effect & analysis.** Only fine-tuning NP's parameter yields two key benefits:
* **Data efficient.** Table 6 and Line 281-285 in manuscript showcase NPD, i-BAU, and ANP performance using few clean data (500, 250, 50 samples). NPD still performs well with few clean data while i-BAU performs much worse. Thus, NPD is much more data efficient.
* **Computationally efficient.** Table 7 in manuscript and the following **Table 3** show the running time of different defense methods on two datasets. It shows the computational efficiency of NPD.

**In summary, above three innovative components distinguish our NPD significantly from existing related methods, and make critical contributions to the superior defense performance of NPD on both effectiveness and efficiency**.

**Table 3: Runnign Time (sec.) in comparison with state-of-the-art defenses with 2500 images on PreAct-ResNet18.**
|Defense|FP|NAD|NC|ANP|i-BAU|NPD|
:-:|:-:|:-:|:-:|:-:|:-:|:-:
CIFAR-10|1169.01|74.39|896.45|58.75|57.23|55.16
Tiny ImageNet|3357|351|42512|1692|887|332

---

### Decision · Program_Chairs · 2023-09-21

**Decision:**

Accept (poster)

**Comment:**

The paper proposes a lightweight and effective backdoor defense by inserting a trainable neural network layer. The layer can remove backdoor behavior by filtering out poisoned features. The method has been shown to be effective via experiments. Questions were raised regarding difference from previous methods and defense power against adaptive attacks. These concerns were addressed reasonably during rebuttal.